# RECURRENT ACTION TRANSFORMER WITH MEMORY

## ABSTRACT

Recently, the use of transformers in offline reinforcement learning has become a rapidly developing area. This is due to their ability to treat the agent's trajectory in the environment as a sequence, thereby reducing the policy learning problem to sequence modeling. In environments where the agent's decisions depend on past events (POMDPs), it is essential to capture both the event itself and the decision point in the context of the model. However, the quadratic complexity of the attention mechanism limits the potential for context expansion. One solution to this problem is to extend transformers with memory mechanisms. This paper proposes a Recurrent Action Transformer with Memory (RATE), a novel model architecture that incorporates a recurrent memory mechanism designed to regulate information retention. To evaluate our model, we conducted extensive experiments on memory-intensive environments (ViZDoom-Two-Colors, T-Maze, Memory Maze, Minigrid-Memory), classic Atari games, and MuJoCo control environments. The results show that using memory can significantly improve performance in memory-intensive environments, while maintaining or improving results in classic environments. We believe that our results will stimulate research on memory mechanisms for transformers applicable to offline reinforcement learning. The code is available at https://anonymous.4open.science/r/RATE-B01F/.

## 1 INTRODUCTION

Transformers (Vaswani et al., 2017), originally developed for Natural Language Processing (NLP), perform well in Reinforcement Learning (RL) (Agarwal et al., 2023; Li et al., 2023): online RL (Parisotto et al., 2020; Esslinger et al., 2022; Melo, 2022; Team et al., 2023), offline RL (Chen et al., 2021; Lee et al., 2022; Jiang et al., 2023), and model-based RL (Chen et al., 2022; Micheli et al., 2023; Robine et al., 2023), including solving the credit assignment problem and working in memory-intensive environments (Chen et al., 2021; Ni et al., 2023; Grigsby et al., 2024), provided that the entire trajectory fits within the model context. However, transformers struggle with long sequences due to quadratic attention complexity, limiting their use in long inference tasks.

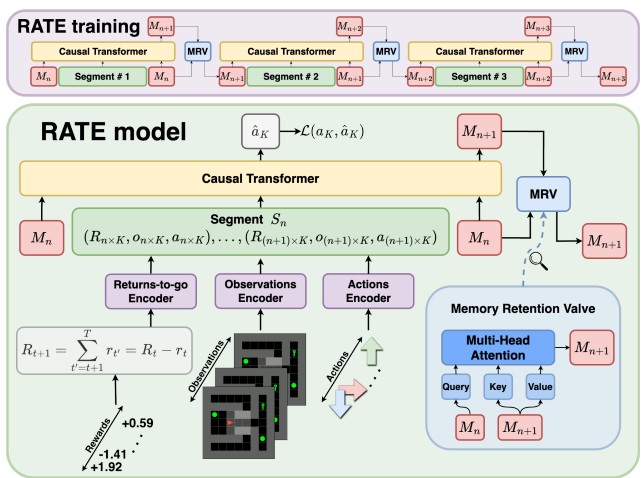

Figure 1: Recurrent Action Transformer with Memory (RATE). $R$ – returns-to-go, $o$ – observations, $a$ – actions, $M_n$ – segment's $S_n$ memory embeddings.

Several approaches attempt to increase the context size (Dai et al., 2019; Bulatov et al., 2022; Ding et al., 2023), but such models may become unstable when trained on long sequences (Zhang et al., 2022), or use a specific sparse attention mechanism that is unsuitable for non-NLP tasks (Beltagy et al., 2020; Zaheer et al., 2020; Ding et al., 2023). Memory mechanisms offer a promising solution to account for past information without increasing context size. Our work explores memory in transformers for RL, inspired by NLP results (Dai et al., 2019; Bulatov et al., 2022). The RL setting differs from NLP in the processing of

input sequences, requiring specialized encoders for observations, rewards, and actions, and is also characterized by significant sparsity in some tasks.

In RL memory has two senses. One is using past information within an episode to make decisions (Lampinen et al., 2021; Ni et al., 2023). The other is transferring experience from one environment to another (Melo, 2022; Kang et al., 2023; Team et al., 2023), improving generalizability, sample efficiency, and solving Meta-RL (Duan et al., 2016; Wang et al., 2016) tasks. Our work focuses on the first case (Ni et al., 2023): using past information to make decisions within the same episode.

In this paper, we propose the **Recurrent Action Transformer with Memory** (**RATE**, Figure 1), a model that uses several memory mechanisms: memory embeddings, caching of previous hidden states of previous tokens, and **Memory Retention Valve** (**MRV**). We empirically show that memory mechanisms effectively preserve information from previous steps, allowing the model to use past information when making decisions in the present. MRV is designed to control the process of updating memory embeddings and prevent the loss of important information when processing long sequences, thus enabling the processing of highly sparse tasks. To evaluate the memory mechanisms, we perform extensive experiments in memory-intensive environments: ViZDoom-Two-Colors (Sorokin et al., 2022), Memory Maze (Pasukonis et al., 2022), Minigrid-Memory (Chevalier-Boisvert et al., 2023), and Passive T-Maze (Ni et al., 2023), as well as on standard RL benchmarks: Atari (Bellemare et al., 2013) and MuJoCo (Fu et al., 2021). We also study the impact of memory on the performance of the proposed model.

The proposed model interpolates and extrapolates well outside the transformer context, is able to retain important information for a long time when operating in highly sparse environments, and allows to compensate the effect of bias in the training data.

Our contribution can be summarized as follows:

1. We propose the Recurrent Action Transformer with Memory (RATE), a transformer model for offline RL that makes use of memory mechanisms: memory embeddings, caching of hidden states of previous tokens, and the Memory Retention Valve (MRV). The proposed MRV is based on the cross-attention architecture and is designed to prevent information loss from memory embeddings and significantly improve the performance of RATE in memory-intensive environments with sparse structure (see section 3).

2. We show that RATE significantly outperforms strong baselines with and without memory mechanisms in memory-intensive environments, including ViZDoom Two-Colors, Memory Maze, Minigrid-Memory, and T-Maze (see subsection 4.2).

3. We demonstrate that RATE achieves better or comparable results in classic Atari games and MuJoCo control tasks, demonstrating that the proposed model is suitable for different types of tasks and emphasizing its universality (see subsection 4.2).

## 2 BACKGROUND

### 2.1 OFFLINE REINFORCEMENT LEARNING

In RL (Sutton & Barto, 2018), we assume that the task can be described by a Markov Decision Process (MDP) as a tuple $\langle \mathcal{S}, \mathcal{A}, \mathcal{P}, \mathcal{R} \rangle$. The process consists of states $s \in \mathcal{S}$, actions $a \in \mathcal{A}$, a state transition function $\mathcal{P}(s'|s, a)$, and an immediate reward function $r = \mathcal{R}(s, a)$. The states are assumed to have the Markov Property, that is $\mathbb{P}(s_{t+1}|s_t) = \mathbb{P}(s_{t+1}|s_1, \ldots, s_t)$. Given a timestep $t$, we use $r_t = R(s_t, a_t)$ to denote the immediate reward that the agent receives at state $s_t$ performing action $a_t$ at that timestep. We describe trajectory $\tau$ of length $T$ as a sequence of states $s_i$, actions $a_i$, and immediate rewards $r_i$: $\tau = (s_0, a_0, r_0, s_1, a_1, r_1, \ldots, s_{T-1}, a_{T-1}, r_{T-1})$. We denote return-to-go (Chen et al., 2021) $R_t$ of trajectory $\tau$ at the timestep $t$ as a sum of future rewards from the timestep $t$ to the end of trajectory: $R_t = \sum_{t'=t}^{T-1} r_{t'}$. The goal of a RL agent is to learn policy $\pi$ that maximizes the expected return. In online RL, the trajectories used to train an agent are obtained iteratively as the agent interacts with the environment. In offline RL, the agent does not interact with the environment during training. A fixed set of trajectories collected by an arbitrary policy is used for training. Although such a setting is more difficult because it does not allow additional exploration of

the environment or generation of new trajectories, it is preferable for tasks where interaction with the environment is costly or risky, such as in robotics.

## 2.2 PARTIALLY OBSERVABLE MARKOV DECISION PROCESS

In the real world, there are frequent situations where the full state of the environment is not available to the agent, so the Markov Property is violated, and the agent is said to receive observations instead of state as input. In this case, observations are no longer sufficient statistics of the past to make a decision in the current step. An example would be a robot navigating an environment based on a camera image or a situation in which a decision must be made based on information from the past that is not available in the current observation. A Partially Observable Markov Decision Process (POMDP) is used in such cases. POMDP is a generalization of the MDP and is written as $\langle \mathcal{S}, \mathcal{A}, \mathcal{O}, \mathcal{P}, \mathcal{R}, \mathcal{Z} \rangle$, where $o \in \mathcal{O}$ – observations, $s \in \mathcal{S}$ – states, $a \in \mathcal{A}$ – actions, $r = \mathcal{R}(s, a)$ – immediate reward function, $\mathcal{P}(s'|s, a)$ – state transition function and $\mathcal{Z}$ is an observation function $\mathcal{Z}_{s'o}^a = P(O = o_{t+1}|S_{t+1} = s', A_t = a)$. To work successfully in such environments, mechanisms such as memory are needed to allow the use of information from the past (Parisotto et al., 2020; Lampinen et al., 2021; Esslinger et al., 2022). In our work, we propose an approach of adding memory to the agent in the form of memory embeddings, caching of hidden states of previous tokens, and MRV. In this paper we consider the setting of an offline model-free RL, where learning is formulated as a sequence modeling problem (Chen et al., 2021).

## 3 RECURRENT ACTION TRANSFORMER WITH MEMORY

In this paper, we introduce a new architecture, Recurrent Action Transformer with Memory (RATE), in which we utilized recurrently trained memory embeddings (Bulatov et al., 2022) and caching of hidden states of previous tokens (Dai et al., 2019) to add memory, and Memory Retention Valve (MRV) to control information leakage from memory embeddings, allowing sparse sequences to be processed. The architecture of RATE is shown in Figure 1.

The RATE scheme is outlined in Algorithm 1. In the initial phase, we obtain embeddings $\tilde{R}, \tilde{o}$ and $\tilde{a}$ from the returns-to-go $R$, observations $o$, and actions $a$, respectively, using the corresponding encoders from Table 10. We then generate a trajectory $\tau_{0:T-1}$ consisting of triplets of these embeddings according to the technique described in the Decision Transformer (DT) (Chen et al., 2021) paper (Algorithm 3).

Next, the trajectory $\tau_{0:T-1}$ is divided into $N = T//K$ segments $S_n \in \mathbb{R}^{3K \times d}$, $n \in [0, N-1]$, each consisting of $K$ triplets, where $K$ is the context length and $d$ is the model dimension. To each segment $S_n$, memory embeddings

---

**Algorithm 1** Recurrent Action Transformer with Memory

**Input**: $R \in \mathbb{R}^T, o \in \mathbb{R}^{d_o \times T}, a \in \mathbb{R}^T$
**Parameters**: $M \in \mathbb{R}^{m \times d}$

1: $\tilde{R} \in \mathbb{R}^{T \times d} \leftarrow \texttt{Encoder}_R(R)$
$\quad \tilde{o} \in \mathbb{R}^{T \times d} \leftarrow \texttt{Encoder}_o(o)$
$\quad \tilde{a} \in \mathbb{R}^{T \times d} \leftarrow \texttt{Encoder}_a(a)$
2: $\tau_{0:T-1} \leftarrow \{(\tilde{R}_0, \tilde{o}_0, \tilde{a}_0), \ldots, (\tilde{R}_{T-1}, \tilde{o}_{T-1}, \tilde{a}_{T-1})\}$
3: $M \leftarrow M_0 \sim \mathcal{N}(0, 1)$
4: **for** n **in range** $[0, T//K - 1]$ **do**
5: $\quad S_n \in \mathbb{R}^{3K \times d} \leftarrow \tau_{n \times K:(n+1) \times K}$
6: $\quad \tilde{S}_n \in \mathbb{R}^{(3K+2m) \times d} \leftarrow \texttt{concat}(M_n, S_n, M_n)$
7: $\quad \hat{a}_n, M_{n+1} \leftarrow \texttt{Transformer}(\tilde{S}_n)$
8: $\quad M_{n+1} \leftarrow \texttt{MRV}(M_n, M_{n+1})$
$\quad$ **Output**: $\hat{a}_n \rightarrow \mathcal{L}(a_n, \hat{a}_n), M_{n+1}$
9: **end for**

---

**Algorithm 2** Memory Retention Valve

**Input**: $M_n, M_{n+1} \in \mathbb{R}^{m \times d}$
**Parameters**: $\mathbf{W}_Q^h, \mathbf{W}_K^h, \mathbf{W}_V^h \in \mathbb{R}^{d_h \times d}, \mathbf{W}_M \in \mathbb{R}^{d \times d}$

1: $\mathbf{Q}_h \leftarrow M_n \mathbf{W}_Q^h{}^T$
2: $\mathbf{K}_h \leftarrow M_{n+1} \mathbf{W}_K^h{}^T$
3: $\mathbf{V}_h \leftarrow M_{n+1} \mathbf{W}_V^h{}^T$
4: $M_{n+1}^h \leftarrow \texttt{softmax}(\frac{\mathbf{Q}_h \mathbf{K}_h^T}{\sqrt{d}})\mathbf{V}_h$
5: $M_{n+1} \leftarrow \texttt{concat}(M_{n+1}^0, \ldots, M_{n+1}^h)$
6: $M_{n+1} \leftarrow M_{n+1} \mathbf{W}_M^T$
**Output**: $M_{n+1}$

---

$M_n \in \mathbb{R}^{m \times d}$ are concatenated at the beginning and at the end, forming a $\tilde{S}_n \in \mathbb{R}^{(3K+2m) \times d}$, where $m$ is the number of memory embeddings. These segments $\tilde{S}_n$ are then fed into the transformer and the output is the predicted actions $\hat{a}_n = \hat{a}_{n \times K:(n+1) \times K}$, which are used to compute loss, and new memory embeddings $M_{n+1}$, which are then processed by the MRV block and transmitted to the next segment $S_{n+1}$.

The MRV scheme is presented in Algorithm 2 and is based on the cross-attention basis. After the transformer processes the segment $S_n$, the previous memory embeddings $M_n$ and the new memory embeddings $M_{n+1}$ obtained at the output of the transformer are fed to the input of the MRV block. Next, $M_n$ are multiplied by the query matrix $\mathbf{W}_Q^h$, and $M_{n+1}$ are multiplied by the key $\mathbf{W}_K^h$ and value $\mathbf{W}_V^h$ matrix for each of the attention heads $h$. Then the attention scores are calculated using the softmax() function, the results for each of all attention heads are concatenated and after linear transformation using the matrix $\mathbf{W}_M^T$ the final memory tokens $M_{n+1}$ are obtained at the output, which is passed to the next segment.

Unlike the DT learning process, where random fragments of length $K$ are cut from trajectories, we process segments of length $K$ sequentially, which allows us to capture all information in the processed trajectory. Thus, using memory mechanisms, we are able to process sequences of length $K_{eff} = K \times N$, where $K_{eff}$ is the effective context (Bulatov et al., 2022).

## 4 EXPERIMENTAL EVALUATION

We designed experiments to accomplish two primary objectives: (a) to demonstrate the advantage of our RATE model in memory-intensive environments (T-Maze (Ni et al., 2023), ViZDoom-Two-Colors (Sorokin et al., 2022), Memory Maze (Pasukonis et al., 2022), Minigrid-Memory (Chevalier-Boisvert et al., 2023)), and (b) to investigate the effectiveness of the proposed model in classical MDPs to demonstrate its versatility (Atari (Bellemare et al., 2013) and MuJoCo (Fu et al., 2021)).

For comparison with RATE, we chose DT (Chen et al., 2021) as the main baseline, and adapted the memory-augmented architectures Recurrent Memory Transformer (RMT) (Bulatov et al., 2022) and Transformer-XL (TrXL) (Dai et al., 2019) developed for NLP tasks to the RL domain. Information about the environments used can be found in Table 7.

### 4.1 MEMORY-INTENSIVE ENVIRONMENTS

To test RATE memory mechanisms, we use memory-intensive environments Figure 2, i.e., environments where the agent requires memory to operate successfully. A brief description of these environments is presented below, and a full description and data collection methodology can be found in the Appendix B.

1. **ViZDoom-Two-Colors** (Sorokin et al., 2022) – an agent in an acid-filled room observes a quickly disappearing green or red pillar. To stay alive, the agent must recall the pillar's color and gather items of the same color.

2. **T-Maze** (Ni et al., 2023) – an agent navigates a T-shaped corridor, receiving a clue at the start about which direction to turn at the end of the corridor. The task tests memory in a prolonged sparse reward environment (the agent receives a reward only at the end).

3. **Memory Maze** (Pasukonis et al., 2022) – an agent navigates a maze, seeking objects matching the color of its view frame. The frame color changes after each successful find. The goal is to collect the most matching objects within a time limit.

4. **Minigrid-Memory** (Chevalier-Boisvert et al., 2023) – a similar task to T-Maze, but with different observation spaces and reward functions (see Appendix B, Table 7). Another important difference is that the agent appears at a random point at the beginning of the episode, not at the beginning of the corridor. Thus, in the case of Minigrid-Memory, it is necessary to reach that clue first (in T-Maze it is a memory problem, in Minigrid-Memory it is a memory and credit assignment problem (Ni et al., 2023)).

In the experiments, the same hyperparameters presented in Table 8 were used for RATE, DT, RMT, and TrXL to simplify the comparison. The context length $K$ and the number of segments $N$ were chosen so that the effective context $K_{eff} = K \times N$ for RATE, RMT, and TrXL covers important events in memory-intensive environments during training. In turn, since DT has no memory mechanisms, for it $K = K_{eff}$. Thus, the context length $K$ for RATE, RMT and TrXL is less than the context length for DT by a factor of $N$, but all models process the same parts of trajectories. More information about the training procedure for each environment can be found in Appendix D.

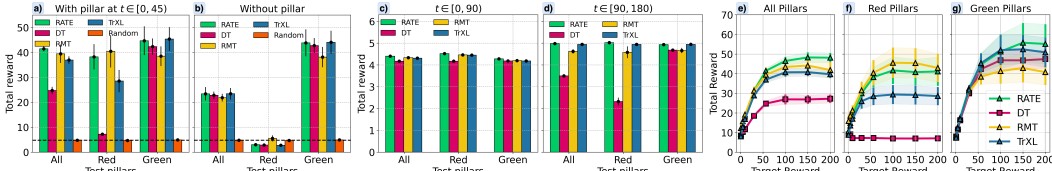

Figure 2: Memory-intensive environments with different observation spaces and reward functions used to test the performance of the memory mechanism in the RATE model.

Figure 3: Results for the ViZDoom-Two-Colors: with (**a**) and without (**b**) pillar in the first 45 steps of the episode; calculated at environment steps $0 - 89$ (**c**) and $90 - 179$ (**d**); depending on the return-to-go (**e, f, g**). Pillar disappears after first 45 steps, $K_{eff} = 90$.

## 4.2 EXPERIMENTAL RESULTS

In this section, the main experimental results for each of the environments are presented in the corresponding paragraphs. Additional results can be found in the Appendix F. For each of the experiments, the same techniques were used for all models to obtain the results presented in the Appendix E. Unless otherwise indicated, all baselines were trained from scratch.

**ViZDoom-Two-Colors.** The dataset for this environment was collected using a pre-trained Advantage Actor Critic (A2C) (Beeching et al., 2019), which has a slight bias in favor of selecting green items even if red items are required. However, the dataset is balanced by the pillars colors. Figure 3 (a) shows the inference results on all pillars, separately only on red pillars and separately only on green pillars. As illustrated in the Figure 3 (a), for inference with the presence of a disappearing pillar at the beginning of the episode, all baselines have an average total reward for inference on green pillars greater than for inference on red pillars.

To prove that this is not due to the peculiarities of the algorithms, but to the presence of bias in the data, we performed an additional inference without a pillar at the beginning of the episode, demonstrating the ability of all baselines to collect exclusively green items. As can be seen from Figure 3 (b), DT learns the distribution of training data and is unable to remember the pillar color. In turn, baselines with memory mechanisms such as RATE, RMT and TrXL are successful in this task. The poor performance of baselines with memory on red pillars without the pillar at the episode beginning proves that it is the color of the pillar that they remember.

This conclusion of DT's inability to use information out of context as opposed to RATE is supported by the experimental results presented in Figure 3 (c, d), which illustrates the inference results for the first 90 steps, where the pillar are entirely captured in the effective context, and the subsequent 90 steps, where the pillar color information begins to disappear from the effective context (the context window moves as a sliding window). As a result, there is a drop in total reward for red-pillar environments by almost a factor of two, indicating DT's inability to memorize information to use it out of context. In turn, for RATE, RMT, and TrXL, the values of total reward in the first and second cases are almost unchanged, indicating their ability to utilize information outside of the current context window.

Figure 3 (e, f, g) demonstrates the dependence of model performance on the return-to-go. In this paper, we used an empirical estimate of the target reward as the average of the top-10 total rewards in the training dataset. As can be seen, RATE not only significantly outperforms DT, but also outperforms the memory-augmented models RMT and TrXL.

Furthermore, Table 1 demonstrates that RATE outperforms not only transformer-based models (RMT, TrXL), but also recurrent baselines, which forget the pillars color fairly quickly and start to collecting green items like DT.

**T-Maze.** The Figure 4 shows the inference results for the T-Maze. To validate an agent's long-term memory capabilities, we performed training on trajectories of length 90 and inference on corridors of size $30 - 900$, i.e. much larger than the effective context $K_{eff} = 90$ of models. The Figure 4 demonstrates that DT's ability to solve the task is limited by the length of the corridors in the training data. Specifically, DT-3, trained on trajectories of length $3 \times 30 = 90$ with $K = K_{eff} = 90$, exhibits a significant drop in performance (demonstrating the performance of the persistent agent, i.e. it successfully reaches the junction, but at the junction it turns one way regardless of the clue) when tasked with inference on corridors exceeding 90 in length.

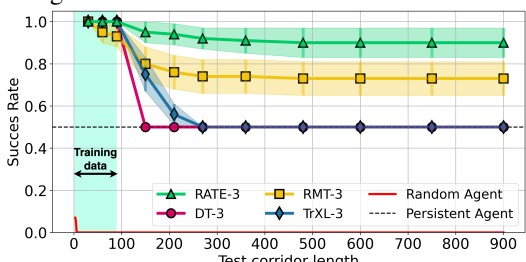

In turn, RATE-3, RMT-3, and TrXL-3 (trained on $3 \times 30 = 90$ steps) perform significantly better at inference corridor lengths longer than the model saw during training. Moreover, RATE-3 outperforms other memory-augmented baselines, indicating its ability to perform effectively in sparse environments. This confirms the ability of the RATE model to successfully memorize important information and retain it through a long time.

Additionally, Table 1 compares RATE with recurrent baselines. The results indicate that these baselines, unlike RATE, cannot handle sparse information, as shown by SR $= 0.5$.

Figure 4: Results for the T-Maze environment with $K_{eff} = 30 \times 3 = 90$. The notations are represented as MODEL-N, where N is the number of segments into which the trajectories are divided. Persistent agent — an agent that always reaches the end of the corridor but selects randomly the up or down action at the turn.

Table 1: Comparison of Transformer (DT), RNN (Decision LSTM (DLSTM) (Siebenborn et al., 2022), Decision GRU (DGRU)) and SSM (Decision Mamba (DMamba) (Ota, 2024)) models with RATE in memory-intensive environments. The results indicate the inability of the RNN and SSM models to train successfully on trajectories of length $90 \times 3 = 270$ tokens, unlike RATE. SR – Success Rate. DGRU is obtained by replacing the LSTM block with the GRU block (Chung et al., 2014) in DLSTM. $^{\dagger}$ $K = 90$ ($K_{eff} = 3 \times 30 = 90$ for RATE).

| | **T-Maze** | | | | | |
|---|---|---|---|---|---|---|
| | **Random** | **DLSTM** | **DGRU** | **DMamba** | **DT** | **RATE** (ours) |
| **SR** ($K = T = 9$) | 0.0 | **1.0** | **1.0** | **1.0** | **1.0** | **1.0** |
| **SR** ($K = T = 30$) | 0.0 | 0.6 | **1.0** | **1.0** | **1.0** | **1.0** |
| **SR** ($K = T = 90$) | 0.0 | $0.5 \pm 0.0$ | $0.5 \pm 0.0$ | $0.5 \pm 0.0$ | $1.0 \pm 0.0$ | $1.0 \pm 0.0$ |
| | **ViZDoom-Two-Colors**$^{\dagger}$ | | | | | |
| **Reward[Total]** | 4.82 | $13.1 \pm 0.6$ | $12.9 \pm 0.2$ | $26.9 \pm 1.9$ | $24.8 \pm 1.4$ | $\mathbf{41.5 \pm 1.0}$ |
| **Reward[Reds]** | 4.66 | $8.8 \pm 0.7$ | $9.4 \pm 0.5$ | $6.9 \pm 0.4$ | $7.2 \pm 0.4$ | $\mathbf{38.2 \pm 5.1}$ |
| **Reward[Greens]** | 4.98 | $17.5 \pm 1.6$ | $16.3 \pm 0.8$ | $\mathbf{46.9 \pm 4.2}$ | $42.3 \pm 3.3$ | $\mathbf{44.7 \pm 5.8}$ |

**Minigrid-Memory.** The Figure 5 shows the results for the Minigrid-Memory. Training was conducted on grids of size 31x31, inference was conducted on grids of size $11x11 - 91x91$. Unlike the previously discussed T-Maze, the credit assignment problem is also addressed here, since the agent first has to reach the oracle and find out which object to turn towards in the future.

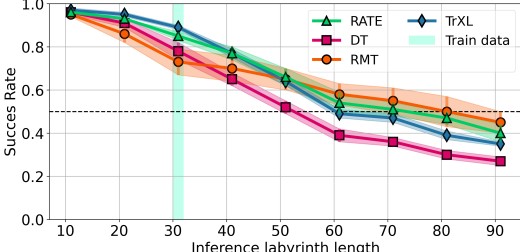

Figure 5: Results for the Minigrid-Memory environment, $K_{eff} = 10 \times 3 = 30$.

The results show that TrXL performs better than the other baselines on grids of size smaller or equal to those used in training, and worse than the other baselines on larger grids, that is, it interpolates well and extrapolates poorly. In turn, for RMT we observe exactly the opposite situation: RMT interpolates poorly and extrapolates well. RATE performs slightly worse than TrXL but better than RMT on small grid sizes, and slightly worse than RMT but better than TrXL on large grid sizes, but on average has interpolation and extrapolation abilities better than RMT and TrXL individually, as well as greater stability.

**Memory Maze.** Table 2 shows the results of comparing RATE with other basic baselines with and without memory. The results show that RATE is able to memorize the implicit information like maze structure more efficiently, which is reflected in a higher average reward per episode.

Table 2: Results for the Memory Maze 9x9 environment.

| | DT (Chen et al., 2021) | RMT (Bulatov et al., 2022) | TrXL (Dai et al., 2019) | RATE |
|---|---|---|---|---|
| Reward | $6.83 \pm 0.51$ | $7.27 \pm 0.21$ | $7.12 \pm 0.24$ | $\mathbf{7.64 \pm 0.41}$ |

**Atari and MuJoCo.** The results for Atari and MuJoCo are presented in Table 3 and Table 4. Results for Decision Mamba (DMamba) (Ota, 2024) and Mamba as Decision Maker (MambaDM) (Cao et al., 2024) are from the corresponding papers. A more detailed description of the results obtained can be found in the Appendix D. The results demonstrate that RATE not only performs as well as the algorithms specifically designed for offline RL, but in many cases outperforms them in classical environments that do not require memory, which indicates the versatility of the model.

Table 3: Raw scores for Atari games. Green – top-1 result, light green – top-2 result within the standard deviation.

| Environment | CQL (Kumar et al., 2020) | DT (Chen et al., 2021) | DMamba (Ota, 2024) | MambaDM (Cao et al., 2024) | RATE |
|---|---|---|---|---|---|
| Breakout | 62.5 | $76.9 \pm 27.3$ | $70.6 \pm 9.3$ | $106.9 \pm 5.8$ | $111.0 \pm 2.9$ |
| Qbert | 14013.2 | $2215.8 \pm 1523.7$ | $5786.0 \pm 1295.2$ | $10052.5 \pm 1116.5$ | $12486.9 \pm 280.4$ |
| SeaQuest | 782.2 | $1129.3 \pm 189.0$ | $992.1 \pm 57.7$ | $1286.0 \pm 42.0$ | $1037.9 \pm 53.7$ |
| Pong | 18.8 | $17.1 \pm 2.9$ | $1.6 \pm 15.3$ | $18.4 \pm 0.8$ | $18.8 \pm 0.3$ |

Table 4: Scores normalized according to the protocol in Fu et al. (2021) for MuJoCo control tasks. ME – Medium-Expert dataset, M – Medium dataset, MR – Medium-Replay dataset. RATE outperforms DT in 9/9 of the cases. Green – top-1 result, light green – top-2 result within the standard deviation.

| Dataset | Environment | CQL (Kumar et al., 2020) | DT (Chen et al., 2021) | TAP (Jiang et al., 2023) | DMamba (Ota, 2024) | MambaDM (Cao et al., 2024) | RATE |
|---|---|---|---|---|---|---|---|
| ME | HalfCheetah | 91.6 | $86.8 \pm 1.3$ | $91.8 \pm 0.8$ | $91.9 \pm 0.6$ | $86.5 \pm 1.2$ | $87.4 \pm 0.1$ |
| ME | Hopper | 105.4 | $107.6 \pm 1.8$ | $105.5 \pm 1.7$ | $111.1 \pm 0.3$ | $110.5 \pm 0.3$ | $112.5 \pm 0.2$ |
| ME | Walker2d | 108.8 | $108.1 \pm 0.2$ | $107.4 \pm 0.9$ | $108.3 \pm 0.5$ | $108.8 \pm 0.1$ | $108.7 \pm 0.5$ |
| M | HalfCheetah | 44.4 | $42.6 \pm 0.1$ | $45.0 \pm 0.1$ | $42.8 \pm 0.1$ | $42.8 \pm 0.1$ | $43.5 \pm 0.3$ |
| M | Hopper | 58.0 | $67.6 \pm 1.0$ | $63.4 \pm 1.4$ | $83.5 \pm 12.5$ | $85.7 \pm 7.8$ | $77.4 \pm 1.4$ |
| M | Walker2d | 72.5 | $74.0 \pm 1.4$ | $64.9 \pm 2.1$ | $78.2 \pm 0.6$ | $78.2 \pm 0.6$ | $80.7 \pm 0.7$ |
| MR | HalfCheetah | 45.5 | $36.6 \pm 0.8$ | $40.8 \pm 0.6$ | $39.6 \pm 0.1$ | $39.1 \pm 0.1$ | $39.0 \pm 0.6$ |
| MR | Hopper | 95.0 | $82.7 \pm 7.0$ | $87.3 \pm 2.3$ | $82.6 \pm 4.6$ | $86.1 \pm 2.5$ | $83.7 \pm 8.2$ |
| MR | Walker2d | 77.2 | $66.6 \pm 3.0$ | $66.8 \pm 3.1$ | $70.9 \pm 4.3$ | $73.4 \pm 2.6$ | $73.7 \pm 1.4$ |
| | Mean | 77.6 | 74.7 | 74.8 | 78.8 | 79.0 | 78.5 |

## 5 ABLATION STUDY

In this section, we answer the following research questions (RQs) to evaluate the impact of memory on model performance:

1. *"How do the different components of RATE affect model performance in memory-intensive environments?"*— RQ 1.

2. *"What is the upper-bound estimate of the performance of the RATE model?"*— RQ 2.

3. *"Why do you need an MRV and what is its best configuration?"*— RQ 3.

**RQ 1. Investigating the impact of RATE components.** To study the influence of memory embeddings on RATE model performance, we conducted the following experiment: for the RATE model trained for T-Maze (with context $K = 30$ on $N = 3$ segments), we replaced pre-trained memory embeddings $M$ with random noise vectors during inference (see Figure 7).

Using random noise instead of memory embeddings, SR $= 50\%$ in the T-Maze, indicating the agent reaches the junction but turns in only one direction regardless of the clue. Thus, we conclude that clue information is in memory embeddings, while the rest of the actions are shaped by transformer parameters.

During RATE inference in ViZDoom-Two-Colors environment with replacement of different components of memory RATE mechanisms (memory embeddings and cached hidden states) with noise (see Figure 6), it is found that for this environment the caching of hidden

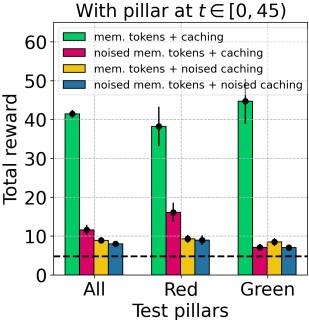

Figure 6: Results of replacing RATE memory tokens and cached hidden states with white noise during inference in ViZDoom-Two-Colors.

states of previous tokens has the largest contribution, because with its noise performance drop is the largest. Thus, in sparse environments, memory embeddings contribute the most, while in continuous environments, caching of hidden states of previous tokens is most impactful.

**RQ 2. Performance upper-bound estimate.** To evaluate the maximum possible performance of the RATE model, we conducted experiments with OracleDT – a DT model whose context is augmented as a pre- and post-fix with a vector $v$ of dimension $1 \times$ `d_model` containing a priori 1-bit information about the environment. Thus, in the T-Maze environment, this information is represented by a clue at the beginning of the episode ($v_i = 0$ if clue = 0 else 1), and in the ViZDoom-Two-Colors environment, it is represented by a column color ($v_i = 0$ if column color = red else 1). A context $S' = \texttt{concat}(v, S, v)$

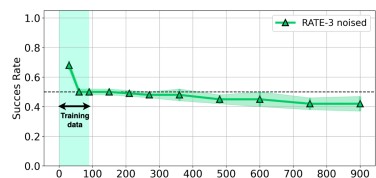

Figure 7: Results of replacing RATE memory tokens with white noise curing inference in T-Maze.

extended in this way can be interpreted as a context $\texttt{concat}(M, S, M)$ with $M$ memory embeddings added, trained perfectly and containing $100\%$ of the important information. Thus, in environments where the a priori information about the environment needed for decision making can be extracted into a given vector $v$, the condition $R[OracleDT] \geq R[RATE] \geq R[DT]$ must be satisfied (see Table 5). This a priori information cannot be extracted from the environment in general, which further emphasizes the advantage of RATE, which is able to automatically extract important information and record it in memory embeddings $M$.

Table 5: Comparison of OracleDT with RATE. OracleDT determines the upper-bound estimate for the maximum reward that can be obtained using RATE in the environment. SR – Success Rate.

| | **T-Maze** | | |
|---|---|---|---|
| | **OracleDT** | **DT** (Chen et al., 2021) | **RATE** |
| | ($K = 90$) | ($K = 90$) | ($K_{eff} = 3 \times 30 = 90$) |
| **SR** ($T = 90$) | $\mathbf{1.0 \pm 0.0}$ | $\mathbf{1.0 \pm 0.0}$ | $\mathbf{1.0 \pm 0.0}$ |
| **SR** ($T = 480$) | $\mathbf{1.0 \pm 0.0}$ | $0.5 \pm 0.0$ | $0.90 \pm 0.07$ |
| **SR** ($T = 900$) | $\mathbf{1.0 \pm 0.0}$ | $0.5 \pm 0.0$ | $0.90 \pm 0.07$ |
| | **ViZDoom-Two-Colors** | | |
| **Reward[Total]** | $\mathbf{56.5 \pm 0.8}$ | $24.8 \pm 1.4$ | $41.5 \pm 1.0$ |
| **Reward[Reds]** | $\mathbf{55.3 \pm 1.6}$ | $7.2 \pm 0.4$ | $38.2 \pm 5.1$ |
| **Reward[Greens]** | $\mathbf{57.2 \pm 0.5}$ | $42.3 \pm 3.3$ | $44.7 \pm 5.8$ |

**RQ 3. Memory Retention Valve architecture ablation.** Without MRV in the T-Maze environment at corridor lengths of $L \gg K$, the performance of the RATE model decreased with each segment processed at inference, resulting in almost SR $= 50\%$ on long trajectories (see Table 6). For example, in the T-Maze task, the important information to be remembered goes into memory embeddings when processing the first segment of the sequence, and then it must be retrieved when making decisions on the last segment. At the same time, due to the recurrent structure of the architecture, memory embeddings continue to be updated during the processing of intermediate segments when no new information needs to be memorized, causing important information from memory embeddings to leak out. To retain important information in memory embeddings, MRV mechanism was added to the architecture. We considered the five different schemes detailed in subsection F.3 to implement MRV:

1. **MRV-CA-1**: cross-attention-based MRV which uses an attention mechanism to control the updating of memory embeddings. The updated memory embeddings $M_{n+1}$ are fed to Query, and the incoming $M_n$ are fed to Key and Value.

2. **MRV-CA-2**: uses the same mechanism as MRV-CA-1 but the incoming memory embeddings $M_n$ are fed to Query, and the updated $M_{n+1}$ are fed to Key and Value.

3. **MRV-G**: gated MRV which uses a gating mechanism similar to the one used in GTrXL (Parisotto et al., 2020).

4. **MRV-GRU**: uses a GRU (Chung et al., 2014) block to process updated $M_n$ with hidden states.

5. **MRV-LSTM**: uses a LSTM (Hochreiter & Schmidhuber, 1997) block to process updated $M_n$ with cached states.

Table 6: Results of ablation study of MRV configuration on T-Maze environment. † – baseline.

| Model ($K_{eff} = 30 \times 5 = 150$) | Inference corridor length | | | |
|---|---|---|---|---|
| | 150 | 360 | 600 | 900 |
| RATE w/o MRV[†] | $1.00 \pm 0.00$ | $0.66 \pm 0.08$ | $0.65 \pm 0.07$ | $0.61 \pm 0.07$ |
| RATE (MRV-CA-2) | $1.00 \pm 0.00$ | $0.95 \pm 0.05$ | $0.90 \pm 0.07$ | $0.90 \pm 0.07$ |
| RATE (MRV-G) | $0.86 \pm 0.07$ | $0.77 \pm 0.08$ | $0.66 \pm 0.07$ | $0.65 \pm 0.08$ |
| RATE (MRV-GRU) | $0.99 \pm 0.01$ | $0.74 \pm 0.07$ | $0.56 \pm 0.11$ | $0.55 \pm 0.12$ |
| RATE (MRV-LSTM) | $0.85 \pm 0.06$ | $0.64 \pm 0.10$ | $0.51 \pm 0.11$ | $0.47 \pm 0.11$ |
| RATE (MRV-CA-1) | $0.51 \pm 0.01$ | $0.51 \pm 0.01$ | $0.49 \pm 0.02$ | $0.49 \pm 0.01$ |

The best results in Table 6 were obtained using cross-attention scheme (MRV-CA-2), in which we fed $M_n$ memory tokens from the transformer input to the query and $M_{n+1}$ memory tokens from the transformer output to the key and value. This configuration is used throughout the work and is denoted simply as MRV. This configuration acts as an effective gating mechanism to prevent the loss of important information in prolonged sparse environments, which is reflected in significantly better results for RATE in the T-Maze environment.

# 6 RELATED WORK

## 6.1 TRANSFORMERS FOR REINFORCEMENT LEARNING

Transformers have found application in various areas of RL (Agarwal et al., 2023; Li et al., 2023): online RL (Parisotto et al., 2020; Lampinen et al., 2021; Esslinger et al., 2022; Melo, 2022; Zheng et al., 2022; Pramanik et al., 2023; Team et al., 2023), offline RL (Chen et al., 2021; Janner et al., 2021; Lee et al., 2022; Reed et al., 2022; Jiang et al., 2023), and model-based RL (Chen et al., 2022; Micheli et al., 2023; Robine et al., 2023). The use of transformers as a general policy for many environments is also being explored (Lee et al., 2022; Melo, 2022; Reed et al., 2022). In our work, we consider the formulation of an offline model-free RL, where learning is formulated as a sequence modeling problem (Chen et al., 2021). Prominent representatives of such models are (Chen et al., 2021; Janner et al., 2021; Lee et al., 2022; Jiang et al., 2023), although planning in latent space (Jiang et al., 2023) is considered, which can be seen as modeling of the environment. Moreover, (Janner et al., 2021; Jiang et al., 2023) are specialized for control tasks with vector observations and do not generalize to environments with observations in the form of images. Therefore, we consider the Decision Transformer (Chen et al., 2021), which has no memory mechanism, as the main baseline for comparison.

## 6.2 RECURRENT NEURAL NETWORKS FOR REINFORCEMENT LEARNING

For long input sequences, recurrent networks may have computational advantages over transformers. The RNNs recurrent unit maintains a hidden state, which is essentially a form of memory that is important for solving POMDPs. In Decision LSTM (DLSTM) (Siebenborn et al., 2022) in DT the transformer is replaced by an LSTM unit (Hochreiter & Schmidhuber, 1997).

## 6.3 STATE SPACE MODELS FOR REINFORCEMENT LEARNING

Recently, State Space Models (SSMs) (Gu et al., 2021) have shown significant success in sequence modeling, particularly in offline RL (Bar-David et al., 2023; Cao et al., 2024; Gu & Dao, 2023; Ota, 2024). In Decision S4 (DS4) (Bar-David et al., 2023), sequence modeling is executed using S4 (Gu et al., 2021) layers within the framework of offline RL, whereas Decision Mamba (DMamba) (Ota, 2024) utilizes the most recent Mamba (Gu & Dao, 2023) sequence model instead of causal self-attention. Mamba Decision Maker (MambaDM) (Cao et al., 2024) integrates the unique features of SSMs to effectively combine local and global features with Global-local fusion Mamba (GLoMa) module.

## 6.4 MEMORY IN TRANSFORMERS

There are many ways to implement the memory mechanism for transformers (Bulatov et al., 2022; Dai et al., 2019; Ding et al., 2020; Lei et al., 2020; Rae et al., 2019; Wu et al., 2020; 2022). In Transformer-XL (TrXL) (Dai et al., 2019), it is proposed to split a long data sequence into segments and to access past segments at the expense of memory, but to ignore very distant segments. This increases the effective length of the context. The Compressive Transformer (Rae et al., 2019) uses

compressed memory, allowing previous versions of memory to be compressed rather than discarded as in TrXL. ERNIE-Doc (Ding et al., 2020) suggests using the retrospective feed mechanism and the enhanced recurrence mechanism. Memformer (Wu et al., 2020) uses external dynamic memory to encode and retrieve past information. MART (Lei et al., 2020) extends this idea by adding a memory update mechanism similar to a recurrent neural network (Cho et al., 2014; Hochreiter & Schmidhuber, 1997). The Memorizing Transformer (Wu et al., 2022) proposes to store the internal representations of past inputs. The Recurrent Memory Transformer (RMT) (Bulatov et al., 2022) includes additional read and write memory tokens at each segment's beginning and end. This method allows the effective context to be expanded to over 1 million tokens (Zhu et al., 2020).

An Adaptive Agent (AdA) (Bauer et al., 2023) uses memory architectures to store and employ information previously acquired by the agent. The default memory architecture is TrXL with normalization before each layer (Parisotto et al., 2020), and the use of gating on the feedforward layers (Shazeer, 2020) to stabilize training. We also use TrXL in our work but refrain from using additional modifications to stabilize training. Another distinctive feature of using a transformer in AdA, as opposed to DT, is that pixel observations, past actions, past rewards, and additional information are not tokenized separately but are combined into a single vector that feeds the transformer. The transformer itself predicts not only actions but also value function values.

## 7 CONCLUSION

In this paper, we propose **Recurrent Action Transformer with Memory** (**RATE**), a transformer model for offline RL that exploits memory mechanisms in the form of memory embeddings and caching of hidden states of previous tokens, and the **Memory Retention Valve** (**MRV**), which controls memory updating and prevents the loss of important information in sparse tasks. In extensive experiments in memory-intensive environments such as ViZDoom-Two-Colors, Memory Maze, Minigrid-Memory, and T-Maze, we have shown that RATE outperforms recurrent and transformer baselines. The proposed model interpolates and extrapolates well outside the transformer context, is able to retain important information for a long time when operating in highly sparse environments, and allows to compensate for the effect of bias in the training data.

We also show that the proposed model achieves better or comparable results to state-of-the-art Mamba-based models in Atari and MuJoCo environments, indicating that RATE is suitable for all tasks: both memory-intensive and not. We have thoroughly investigated the influence of memory mechanisms on the performance of the model and have clearly shown that the model uses them in decision making. This method shows great potential for tackling complex tasks with long sequences, especially in robotics, where training agents on pre-collected data sets is highly advantageous.

**Limitations.** Limitations of the proposed model include its inability to design $K$ and $N$ in memory-intensive environments so that all important events fall into the efficient context of $K_{eff} = K \times N$. In addition, the approach based on dividing trajectories into segments during inference does not allow for memory updates effectively at each step using a sliding window. Also, there are currently no studies of the memory capacity of the proposed model, so the practical amount of information that can be stored remains unknown.

**Reproducibility Statement.** The model description is presented in section 3 (Algorithm 1 and Algorithm 2), the training procedure is presented in Appendix D, the description of the used benchmarks is presented in Appendix B, the hyperparameters are presented in Table 8, and the configurations for displaying the experimental results are presented in Table 9. The results of the hyperparameters tuning for recurrent baselines are presented in Appendix G.

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

## A  DECISION TRANSFORMER

Decision Transformer (DT) (Chen et al., 2021) is an algorithm for offline RL that reduces the RL task to a sequence modeling task. In DT, the scheme of which is presented in Algorithm 3, the trajectory $\tau$ is not divided into segments as in RATE. Instead, random fragments of length $K$ are sampled from the trajectory, since originally this architecture was designed to work only with MDP. The predicted actions $\hat{a}$ are sampled autoregressively.

---

**Algorithm 3** Decision Transformer

**Input**: $R \in \mathbb{R}^{1 \times T}, o \in \mathbb{R}^{d_o \times T}, a \in \mathbb{R}^{1 \times T}$

1: $\tilde{R} \in \mathbb{R}^{T \times d} \leftarrow \texttt{Encoder}_R(R)$
   $\tilde{o} \in \mathbb{R}^{T \times d} \leftarrow \texttt{Encoder}_o(o)$
   $\tilde{a} \in \mathbb{R}^{T \times d} \leftarrow \texttt{Encoder}_a(a)$
2: $\tau_{0..T} \leftarrow \{(\tilde{R}_0, \tilde{o}_0, \tilde{a}_0), \ldots, (\tilde{R}_T, \tilde{o}_T, \tilde{a}_T)\}$
3: $n = \texttt{random}(0, T - K)$
4: $\hat{a}_n \leftarrow \texttt{Transformer}(\tau_{n..n+K})$

**Output**: $\hat{a}_n \rightarrow \mathcal{L}(a_n, \hat{a}_n)$

---

## B  ENVIRONMENTS

### B.1  MEMORY-INTENSIVE ENVIRONMENTS

In this section, we provide an extended description of the environments used in this paper, as well as the methodology used to collect the trajectories. Table 7 summarizes the observations type, rewards type, and actions type for each of the environments considered in this paper.

**ViZDoom-Two-Colors.**  We used a modified ViZDoom-Two-Colors environment from (Sorokin et al., 2022) to assess the model's memory abilities. The agent initially having 100 hit points (HP) is placed in a room without inner walls filled with acid. At each step in the environment, the agent loses a fixed amount of health ($10/32$ HP per step). In the center of the environment, there is a pillar of either green or red color, which disappears after $45$ environment steps. Throughout the environment, objects of two colors (green and red) are generated. When the agent interacts with an object of the same color as the pillar, it gains an increase in health of $+25$ and a reward of $+1$. When the agent interacts with an object of the opposite color, it loses a similar amount of health. The agent receives an additional reward of $+0.02$ for each step it survives. The episode ends when the agent has zero health. Thus, the agent needs to remember the color of the pillar to select items of the correct color, even if the pillar is out of sight or has disappeared. The agent does not receive information about its current health or rewards, as these observations essentially convey the same information as the color of the pillar but persist beyond step $45$.

We collected a dataset of 5000 trajectories of 90 steps in length using a trained A2C (Beeching et al., 2019) agent (an agent trained with a non-disappearing pillar). The average reward for these 90 steps is $4.46$. When collecting trajectories, to ensure that the agent saw the pillar before it disappeared, the agent always appeared facing the pillar in the same place – midway between the pillar and the nearest wall. In order to successfully complete this task, the agent needs to remember the color of the pillar. This environment tests the long-term memory mechanism, since the agent needs to retain information about the pillar for a time much longer than the pillar has been in the environment. Using only short-term memory and, for example, collecting the next item of the same color as the previous collected item, it will not be possible for the agent to survive for a long time, as this policy is extremely unstable. This is due to the fact that in the training dataset the agent occasionally makes a mistake and picks up an object of the opposite color. Thus, irrelevant information about the desired color may enter the transformer context and the agent will start collecting items of an opposite color, which will quickly lead to a failure.

**T-Maze.**  To investigate agent's long-term memory on very long environments (the inference trajectory length is much longer than the effective context length $K_{eff}$) we used a modified version of the T-Maze environment (Ni et al., 2023). The agent's objective in this environment is to navigate from the beginning of the T-shaped maze to the junction and choose the correct direction, based on a signal given at the beginning of the trajectory using four possible actions $a \in \{left, up, right, down\}$. This signal, represented as the $clue$ variable and equals to zero everywhere except the first observation, dictates whether the agent should turn up ($clue = 1$) or down ($clue = -1$). Additionally, a constraint on the episode duration $T = L + 2$, where the maximum duration is determined by the length of the corridor $L$ to the junction, adds complexity to the problem. To address this, a binary flag, represented

as the $flag$ variable, which is equal to $1$ one step before the junction and $0$ otherwise, indicating the arrival of the agent at the junction, is included in the observation vector. Additionally, a noise channel is added to the observation vector, with random integer values from the set $\{-1, 0, +1\}$. The observation vector is thus defined as $o = [y, clue, flag, noise]$, where $y$ represents the vertical coordinate. The reward $r$ is given only at the end of the episode and depends on the correctness of the agent's turn at the junction, being $1$ for a correct turn and $0$ otherwise. This formulation deviates from the traditional Passive T-Maze environment (Ni et al., 2023) (different observations and reward functions) and presents a more intricate set of conditions for the agent to navigate and learn within the given time constraint.

The dataset consists of 2000 of trajectories for each segment of length 30 (i.e. 6000 trajectories for the $K_{eff} = 3 \times 30 = 90$) and consists only of successful episodes. An artificial oracle with a priori information about the environment was used to generate the dataset.

Table 7: Description of observations and reward functions for the considered environments.

| Environment | Obs. type | Rewards | Actions | Obs. info |
|---|---|---|---|---|
| ViZDoom-Two-Colors | Image | Continuous | Discrete | First-person view |
| T-Maze | Vector | Sparse & Discrete | Discrete | Vector |
| Memory Maze | Image | Sparse & Discrete | Discrete | First-person view |
| Minigrid-Memory | Image | Sparse | Discrete | Observes the $3 \times 3$ part of the grid |
| Action Associative Retrieval | Vector | Sparse & Discrete | Discrete | Vector |
| Atari | Image | Sparse & Discrete | Discrete | Observes the full game screen |
| MuJoCo | Vector | Continuous | Continuous | Vector |

**Memory Maze.** In this first-person view 3D environment (Pasukonis et al., 2022), the agent appears in a randomly generated maze containing several objects of different colors at random locations. The agent's task is to find an object of the same color in the maze as the outline around its observation image. After the agent finds an object of the desired color and steps on it, the color of the outline changes and the agent must find another object. The agent receives a $+1$ reward for stepping on the correct object. Otherwise, it receives no reward. The duration of an episode is a fixed number and is equal to 1000. Thus, the agent's task is to find as many objects of the desired color as possible in a limited time. The agent's effectiveness in this environment depends on its ability to memorize the structure of the maze and the location of objects in it in order to find the desired objects faster. Using the Dreamer model (Hafner et al., 2019) to collect dataset of 5000 trajectories only achieved an average award of $4.7$ per episode, i.e., a rather sparse dataset.

**Minigrid-Memory.** Minigrid-Memory (Chevalier-Boisvert et al., 2023) is a 2D grid environment designed to test an agent's long-term memory and credit-assignment (Ni et al., 2023). The environment map is a T-shaped maze with a small room with an object inside it at the beginning of the corridor. The agent appears at a random coordinate in the corridor. The agent's task is to reach the room with the object and memorize it, then reach the junction at the end of the maze and make a turn in the direction where the same object is located as in the room at the beginning of the maze. A reward $r = 1 - 0.9 \times \frac{t}{T}$ is given for success, and $0$ for failure. The episode ends after any agent turns at a junction or after a limited amount of time (95 steps) has elapsed. The agent's observations are limited to a $3 \times 3$ size frame. 10000 trajectories with grid size 31x31 were collected using PPO (Schulman et al., 2017) with TransformerXL (Pleines et al., 2023) with a context length equal to the maximum episode duration.

### B.2 STANDARD BENCHMARKS

**Atari games.** For the Atari game environments (Bellemare et al., 2013), we used the same dataset as in DT, namely the DQN replay dataset with grayscale state images (Agarwal et al., 2020). This

dataset contains 500 thousand of the 50 million steps of an online DQN (Mnih, 2013) agent for each game. We use the following set of games: SeaQuest, Breakout, Pong and Qbert.

**MuJoCo.**   Despite the fact that memory is not required in decision making in control environments like MuJoCo (Fu et al., 2021), we conducted additional experiments in this environment to compare with DT. For the continuous control tasks, we selected a standard MuJoCo locomotion environment and a set of trajectories from the D4RL benchmark (Fu et al., 2021). Since we chose DT and TAP as the main models for comparison on this data, we focused on the environments used in both works (HalfCheetah, Hopper, and Walker). We used three different dataset settings: 1) **Medium** – 1 million timesteps generated by a "medium" policy that achieves about a third of the score of an expert policy; 2) **Medium-Replay** – the replay buffer of an agent trained with the performance of a medium policy (about 200k–400k timesteps in our environments); 3) **Medium-Expert** – 1 million timesteps generated by the medium policy concatenated with 1 million timesteps generated by an expert policy. The scores for the MuJoCo experiments are normalized such that 100 represents an expert policy, following the benchmark protocol outlined in (Fu et al., 2021). The performance metrics for Conservative Q-Learning (CQL) and Trajectory Autoencoding Planner (TAP) are reported from the TAP paper (Jiang et al., 2023), and for DT from the DT paper (Chen et al., 2021), as they use the same dataset and evaluation protocol.

## C    ACTION ASSOCIATIVE RETRIEVAL

As shown in section 4.2, DT has a SR $= 50\%$ for inference at corridor lengths longer than the transformer context length. This is due to the fact that even a DT trained on balanced data has a slight bias in the predicted probability towards one of the two required actions, which leads to the fact that when $t > K$ the agent constantly produces only one action: up or down. In turn, the presence of memory in the agent allows us to combat this problem.

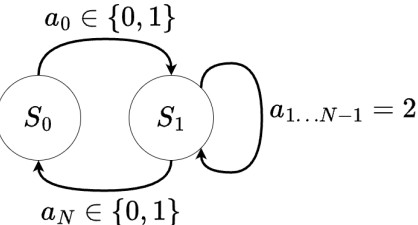

Figure 8: Action Associative Retrieval.

To check how the agent's performance changes during training, we design an **Action Associative Retrieval** (**AAR**) Figure 8 environment.

There are two states in this environment: $S_0$ and $S_1$. The agent appears in state $S_0$ and by performing the action $a_0 \in \{0, 1\}$ moves to state $S_1$. Next, the agent must take $N - 2$ steps to move from state $S_1$ to state $S_1$ by performing action $a = 2$ (no op.). At the end of the episode, the agent must perform the same action that moved it from state $S_0$ to state $S_1$ in order to move from state $S_1$ to state $S_0$. Thus, the action $a \in \{0, 1, 2\}$. Agent observations $o = [state, flag, noise]$, where $state \in \{0, 1\}$ is the index of the current state, $flag \in \{0, 1\}$ is a flag equal to 1 in case the next step requires returning to the initial state and equal to 0 otherwise, $noise \in \{-1, 0, +1\}$ is the noise channel. The agent receives a $+1$ reward if it returns to the initial state $S_0$ by performing the action that took it out from the $S_0$ to the $S_1$, and $-1$ in other cases. The training dataset consists of oracle-generated 6000 trajectories with positive reward.

More formally, we can talk about the presence of memory in an agent when solving AAR (T-Maze-like) tasks under the condition that:

$$\forall t > K : \frac{1}{N_0} \sum_{i=1}^{N_0} p_i(a_t = a^0 | a_0 = a^0) + \frac{1}{N_1} \sum_{i=1}^{N_1} p_i(a_t = a^1 | a_0 = a^1) > 1 \qquad (1)$$

This condition means that if the agent has memory, the sum of the average conditional probabilities over all experiments will be greater than one, i.e., these probabilities are independent of each other. Provided that the sum of these probabilities is less than or equal to one, the agent will choose at best the same target action in most experiments, even if another action is required.

where $a^0, a^1 \in \mathcal{A}$ – two mutually exclusive actions leading to a reward; $t$ is the step at which the final action is required; $N_0, N_1$ are the number of experiments in environments where target action $a_t = a^0$ and $a_t = a^1$, respectively.

In the results Figure 9, the first $1\%$ of training steps was removed because it corresponds to the beginning of the training and is unrepresentative. Blue dots correspond to the beginning of training,

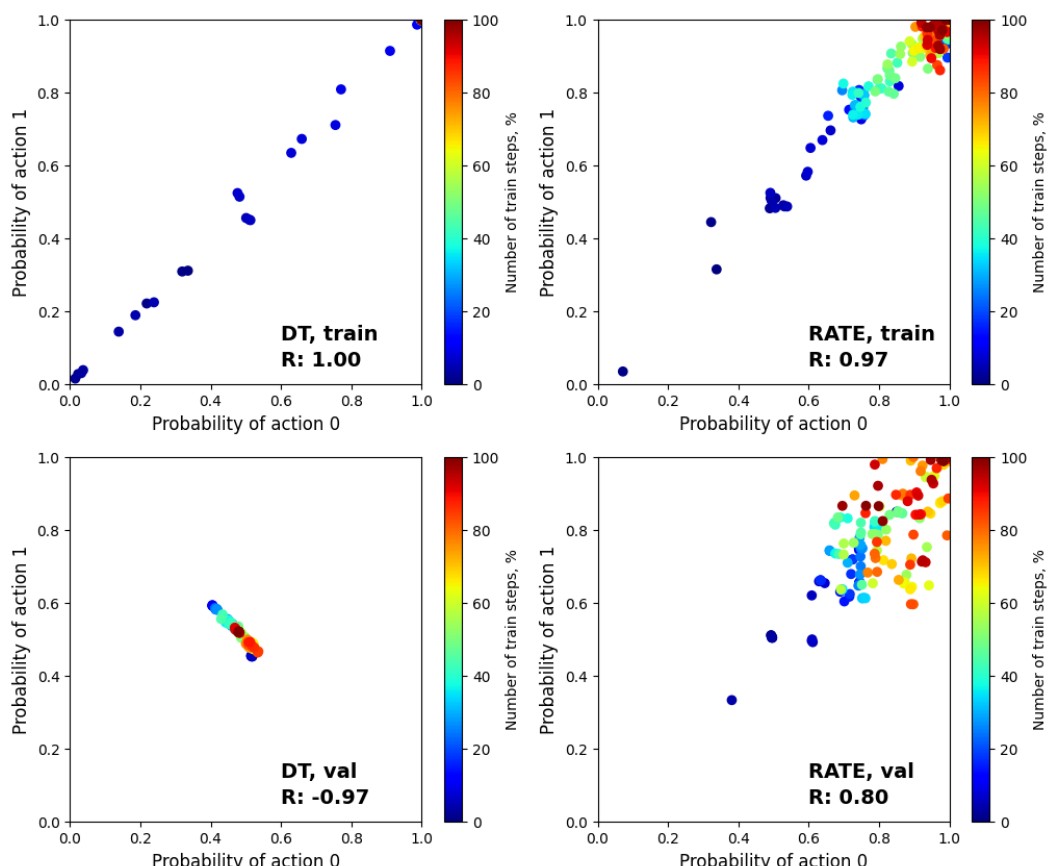

Figure 9: Experimental results with RATE and DT in the AAR environment. The graphs show the 10-runs average results of training on trajectories of length $T = 90$ and validation on trajectories of length $T = 180$, for RATE with $K_{eff} = 3 \times 30 = 90$ and for DT with $K = 90$.

red dots to the end of training. As can be seen from Figure 9, during training, the probabilities $p_i(a_t = a^0 | a_0 = a^0)$ and $p_i(a_t = a^1 | a_0 = a^1)$ on the training trajectories have a strong positive correlation ($R^{DT}_{train} = 1.00$ and $R^{RATE}_{train} = 0.97$), where $R$ – correlation coefficient. This indicates that within-context (effective context) DT and RATE models are able to predict both $a^0$ and $a^1$ actions equally well.

At the same time, during validation, for the RATE model this pattern is preserved – the red points corresponding to the probabilities of choosing actions $a^0$ and $a^1$ are in the upper right part of the graph, positive correlation persists ($R^{RATE}_{val} = 0.80$). On the other hand, in the DT case, the cluster of red dots is skewed toward choosing action $a^1$ and action $a^0$ with equal probabilities equal to $0.5$. Thus, in sum, these probabilities are less or equal to one, as evidenced by a strong negative correlation ($R^{DT}_{val} = -0.97$). The results confirm the inability of DT to generalize on trajectories whose lengths exceed the context length and the ability of RATE to handle such tasks.

## D TRAINING

This section provides additional details on the training process of the baselines considered in the paper. It is important to note that when training RATE in the transformer decoder the feed-forward network block was disabled, because without it on some environments the training results are slightly better. However, other transformer-based baselines were trained with the standard transformer decoder.

**ViZDoom-Two-Colors.** Since the pillar disappears at time $t = 45$, all trajectories start at time $t = 0$ and end at time $t = 90$ so that the information about the color of the pillar is guaranteed to

be used in training. In this experiment, we compared DT with context length $K = 90$ to RATE, RMT, and TrXL models with context length $K = 30$ and partitioning the trajectory into $N = 3$ segments. Thus, when trained, RATE also handles sequences of length 90, since its effective context is $K_{eff} = N \times K = 90$, but only processed subsequences of length $K = 30$.

**T-Maze.** The model names are written in the format MODEL-N, where $N$ is the number of segments of length $K = 30$ into which the training trajectories can be partitioned. Thus, DT-3 was trained on trajectories of length $T \leq 3 \times 30 = 90$ with context length. RATE-3 was trained on similar trajectories as DT-3, but with each trajectory divided into 3 segments, during training, enabling the training of a model with a context length of $K = 30$ on trajectories of length $T = 90$. All the trajectories used in training start from $t = 0$, i.e., from the moment of receiving a clue.

**Memory Maze.** To train RATE, DT, RMT, and TrXL on Memory Maze, the same approach was used as for ViZDoom-Two-Colors environment, except that trajectories were sampled not from $t = 0$ but from $t : \sum_{t'=t}^{t+90} r_{t'} \geq 2$.

As in the ViZDoom-Two-Colors case, training for DT was performed with a context length of $K = 90$ and for RATE, RMT, and TrXL with a context length of $K = 30$ and number of segments $N = 3$, i.e., effective context length $K_{eff} = N \times K = 3 \times 30 = 90$.

**Minigrid-Memory.** To train RATE, DT, RMT, and TrXL in this environment, trajectories were sampled in the same manner as for T-Maze. An environment configuration with a maze of size 31x31 was used as a training configuration. Since the maximum episode duration is 95, training proceeded in the following setting: for DT the context length $K = 30$, for RATE, RMT, and TrXL the context length $K = 10$ and the number of segments $N = 3$. All trajectories, as in T-Maze, are sampled from time $t = 0$.

**Atari and MuJoCo.** When training RATE on Atari games and MuJoCo control tasks, sequences of length $T = 90$ (Atari) and $T = 60$ (MuJoCo) were sampled randomly from the original trajectories in the dataset. These trajectories were then divided into $N = 3$ segments of length $K = 30$ (Atari) and $K = 20$ (MuJoCo), forming an effective context of length $K_{eff} = N \times K = 90$ (60 for MuJoCo).

For Atari, we used the identical experimental design described in the DT paper (Chen et al., 2021). It is worth noting that we presented raw scores for Atari, rather than gamer-normalized scores as described in the DT paper. Table 3 shows the results for Atari environments. RATE outperforms DT significantly in environments like Breakout and Qbert. We attribute this to the observation that, although these environments do not explicitly demand memory, intricate dynamics from the past exert a greater influence on agent behavior than in environments such as SeaQuest. Actions executed in the past notably alter the present state of the environment in Breakout and Qbert, whereas in SeaQuest, such actions hold little significance. For instance, the emergence of enemies and divers in SeaQuest is entirely independent of the agent's prior actions.

For MuJoCo, our findings suggest that the conventional strategy of utilizing return is not suitable for our segment-based scheme. The issue arises during the trajectory, where the agent's return persistently diminishes. However, the true value of the agent's state at the onset and conclusion of the episode could remain unchanged, provided the agent's policy performs consistently well. To rectify this discrepancy, we propose a novel evaluation strategy for MuJoCo tasks. In this approach, each segment commences with the maximum return, simulating the scenario where the agent initiates the trajectory anew. This method effectively mitigates the aforementioned issue, enhancing the accuracy of our evaluation process. Our MuJoCo experiments in Table 4 show that this benefits performance significantly for some environments. Thus, using RATE allowed us to obtain the best metrics for MuJoCo in 4/9 cases compared to the other baselines. RATE also outperforms DT in 9/9 tasks.

## E    RESULTS PRESENTATION

This section provides information on how the presented experimental results were obtained. $N_{runs}$ denotes the number of model runs; $N_{seeds}$ denotes the number of inference episodes with different seeds; sem denotes standard error of the mean, and std denotes standard deviation.

Table 8: RATE hyperparameters for different experiments. ‡ – Leaky ReLU in the Atari.Pong case.

| Hyperparameter | ViZDoom2C / Memory Maze | T-Maze / Minigrid-Memory | Atari | MuJoCo |
|---|---|---|---|---|
| Number of layers | 6 | 8 | 6 | 3 |
| Number of attention heads | 8 | 10 | 8 | 1 |
| Embedding dimension | 128 / 64 | 64 | 128 | 128 |
| Context length K | 30 | 30 / 10 | 30 | 20 |
| Number of segments | 3 | 3 | 3 | 3 |
| Hidden dropout | 0.2 / 0.5 | 0.05 / 0.2 | 0.2 | 0.2 |
| Attention dropout | 0.05 / 0.2 | 0 / 0.05 | 0.05 | 0.05 |
| Number of memory tokens | 5 / 15 | 5 / 15 | 15 | 15 |
| Number of cached tokens (`mem_len`) | 300 / 360 | 0 / 180 | 360 | 2 |
| Max epochs | 100 / 80 | 50 / 250 | 10 | 10 |
| Batch size | 64 | 64 | 128 | 4096 |
| Weight decay | 0.1 | 0.1 | 0.1 | 0.1 |
| Loss function | CE | CE | CE | MSE |
| Optimizer | AdamW | AdamW | AdamW | AdamW |
| MRV activation | ReLU | ReLU | ReLU‡ | ReLU |
| MRV number of attention heads | 2 / 4 | 4 / 1 | 2 | 2 |
| Learning rate | 3e-4 | 3e-4 | 3e-4 | 1e-4 |
| AdamW $(\beta_1, \beta_2)$ | (0.9, 0.95) | (0.9, 0.95) | (0.9, 0.95) | (0.9, 0.95) |

Table 9: Experimental parameters used to present the final results.

| Environment | $N_{runs}$ | $N_{seeds}$ | Metric | Notation |
|---|---|---|---|---|
| ViZDoom-Two-Colors | 6 | 100 | Total reward | mean $\pm$ sem |
| T-Maze | 10 | 100 | Success Rate | mean $\pm$ sem |
| Memory Maze | 3 | 100 | Total reward | mean $\pm$ sem |
| Minigrid-Memory | 3 | 100 | Total reward | mean $\pm$ sem |
| Action Associative Retrieval | 10 | — | Success Rate | mean $\pm$ sem |
| Atari | 3 | 100 | Total reward | mean $\pm$ std |
| MuJoCo | 3 | 100 | Total reward | mean $\pm$ std |

## F  ADDITIONAL ABLATION STUDIES

To determine the optimal hyperparameters associated with memory mechanisms, additional ablation studies were performed in ViZDoom-Two-Colors and T-Maze environments, and the results are presented in Figure 11 and Figure 10 (right). From the ablation studies results, it was found that for environments like ViZDoom-Two-Colors with continuous reward signal and image observations, the best results can be obtained using number of cached memory tokens `mem_len` $= (K \times 3 + 2 \times$ `num_mem_tokens`$) \times N$, where $K$ – context length and $N$ – number of segments.

On the other hand, for environments with sparse events like T-Maze, it has been found that using caching of hidden states of previous tokens (`mem_len` $> 0$) prevents remembering important information. In this case, gating with `n_head_ca` $= 4$ and moderate number of memory tokens `num_mem_tokens` $= 5$ gives the best results (see Figure 10 (right)).

### F.1  ADDITIONAL VIZDOOM-TWO-COLORS ABLATION

The effect of combining of memory tokens with noise is shown in Figure 10 (left). The noise was applied as a convex combination: `memory_tokens` $= (1-\alpha) \times$ `memory_tokens` $+ \alpha \times$ `noise`. With unchanged caching of hidden states of previous tokens at growth of the noise parameter $\alpha$, at first there is a decrease of performance at inference on green pillars (up to $\alpha = 0.5$), and only then a decrease of performance at inference on red pillars. This phenomenon can be explained by the fact that memory embeddings is trained to record mostly information about red pillars, which helps to combat bias in the training data.

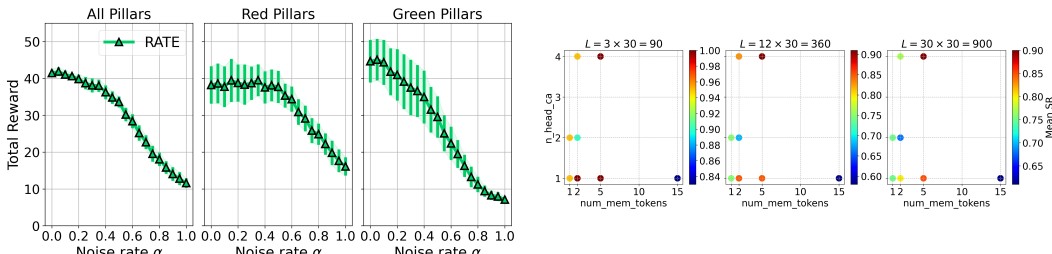

Figure 10: (**left**) Investigating the RATE memory tokens noise effect in the ViZDoom-Two-Colors. (**right**) Results of RATE-3 (trained on corridor lengths ≤ 90) ablation studies in the T-Maze environment. `n_head_ca` – number of MRV attention heads, `num_mem_tokens` – number of memory tokens.

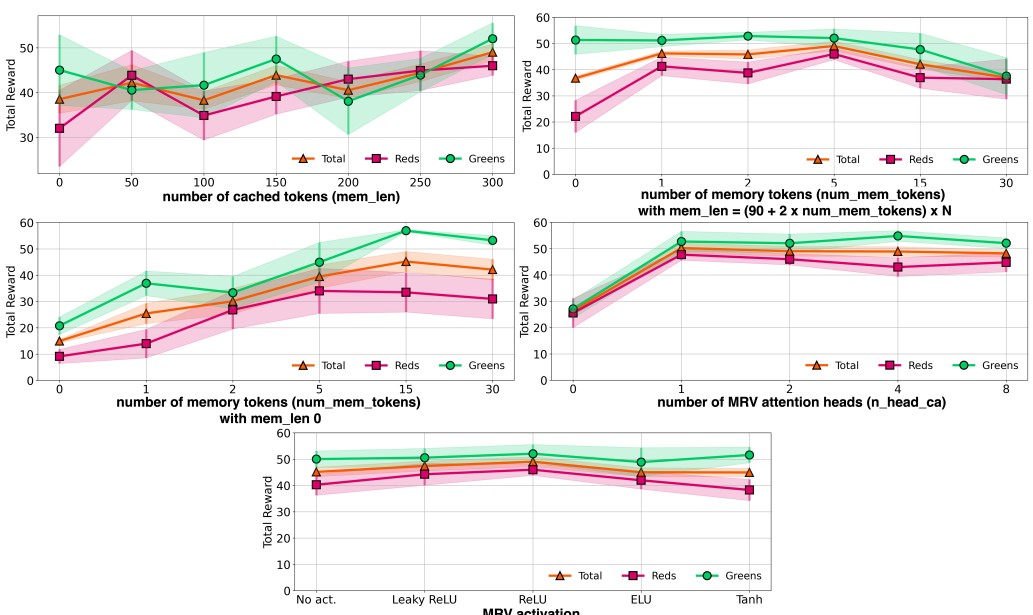

Figure 11: Results of RATE ablation studies in the ViZDoom-Two-Colors environment.

## F.2 CURRICULUM LEARNING

Since in the T-Maze environment, the number of actions at the junction relates to the number of actions when moving straight along the corridor as $\frac{1}{L}$ and tends to $0$ as $L$ increases, there is a significant imbalance in the agent's action distribution, which can cause problems when performing rare class (turning actions) prediction. Theoretically, this situation can be remedied through curriculum learning.

Curriculum learning (CL) is a technique in which a model is trained on examples of increasing difficulty. In this approach, the model is first trained on the set of trajectories $Q_1 = q_1$ of length $K \times 1$, then the trained model is re-trained on the set of trajectories $Q_2 = q_1 \cup q_2$, where the set $q_2$ is formed by trajectories of length $K \times 2$, and so on (in order of increasing complexity of the trajectories). Thus, for the $N$ segments considered during training, the set $Q_N = \bigcup_{i=1}^{N} q_i$ is used.

In the T-Maze environment, DT, RATE, RMT, and TrXL were trained with and without curriculum learning because this approach theoretically produces better results. However, it is important to note that the T-Maze task is successfully solved by the RATE model without using curriculum learning, and even vice versa – its use slightly degraded performance on long corridors. However, with respect to TrXL, the use of CL yielded slightly better results. The work showed that using CL does not achieve significantly better performance on the T-Maze task. The results of using the CL on the

Table 10: RATE encoders for each part of $(R, o, a)$ triplets. We use Embedding layer for encoding discrete actions and Linear for continuous ones. ‡ – channels / kernel sizes / padding.

| Env. | R | O | Conv. configuration‡ | A |
|---|---|---|---|---|
| ViZDoom-Two-Colors | Linear | Conv2D $\times$ 3 | (32, 64, 64) / (8, 4, 3) / 0 | Embedding |
| T-Maze | Linear | Linear | – | Embedding |
| Memory Maze | Linear | Conv2D $\times$ 3 | (32, 64, 64) / (8, 4, 3) / 2 | Embedding |
| Minigrid-Memory | Linear | Conv2D $\times$ 3 | (32, 64, 64) / (8, 4, 3) / 0 | Embedding |
| Atari | Linear | Conv2D $\times$ 3 | (32, 64, 64) / (8, 4, 3) / 0 | Embedding |
| MuJoCo | Linear | Linear | – | Linear |

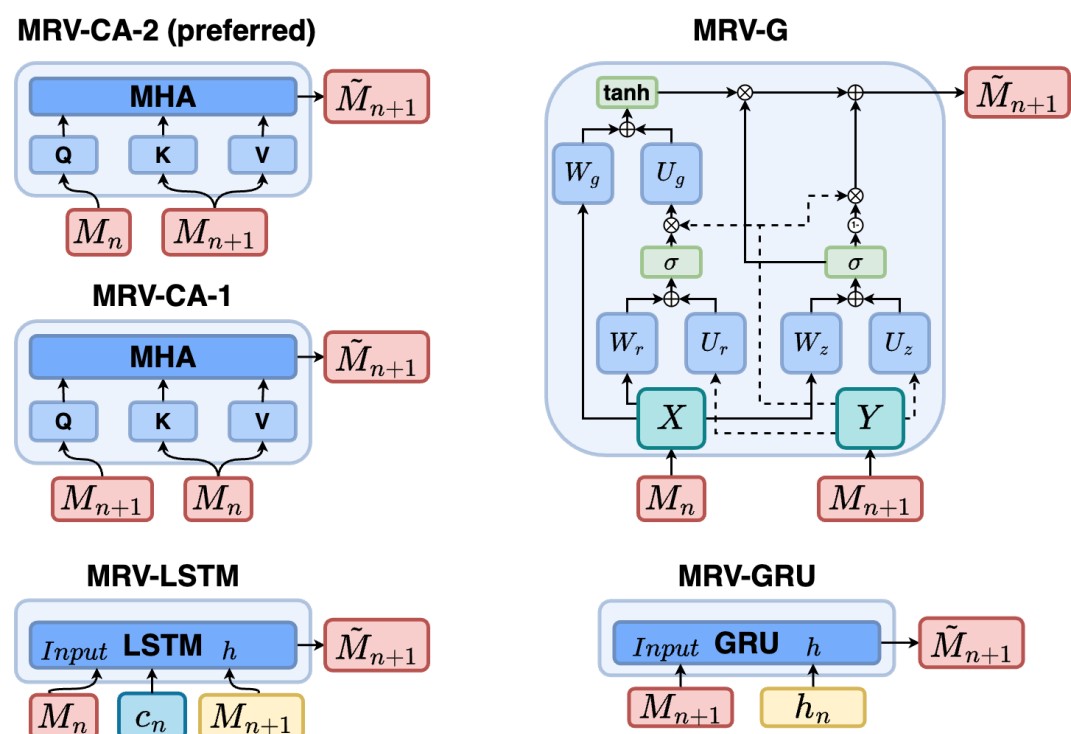

Figure 12: Memory Retention Valve configurations used in the ablation study. **MRV-CA-2**: cross-attention-based MRV which uses an attention mechanism to control the updating of memory embeddings and which is used in the work as the main mechanism. **MRV-CA-1**: uses the same mechanism as MRV-CA-2 but the updated memory embeddings $M_{n+1}$ are fed to Query, and the incoming memory embeddings $M_n$ are fed to Key and Value. **MRV-G**: gated MRV which uses a gating mechanism similar to the one used in Gated Transformer-XL (Parisotto et al., 2020). **MRV-GRU**: uses a GRU (Chung et al., 2014) block to process updated memory embeddings with hidden states. **MRV-LSTM**: uses a LSTM (Hochreiter & Schmidhuber, 1997) block to process updated memory embeddings with cached states.

T-Maze environment are presented in Figure 13 (left), and the results of applying noise to memory embeddings to assess its importance are presented in Figure 13 (right).

### F.3 SUPPLEMENTAL MRV ABLATION

One of the options for implementing the memory tokenization gating mechanism was an approach similar to the one proposed in Gated Transforer-XL (GTrXL) (Parisotto et al., 2020) work. Thus, the MRV-G scheme was inspired by the gating mechanism from GTrXL and implemented as follows:

$$r = \sigma(M_n W_r + M_{n+1} U_r) \tag{2}$$

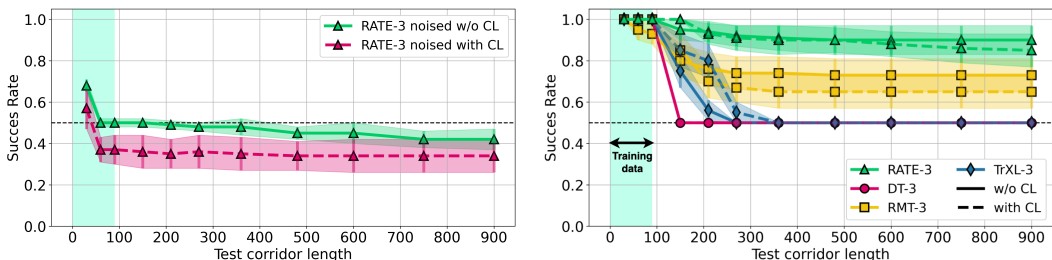

Figure 13: (**left**). Results with and without the use of curriculum learning and (**right**) results of replacing RATE memory tokens with white noise at inference in T-Maze.

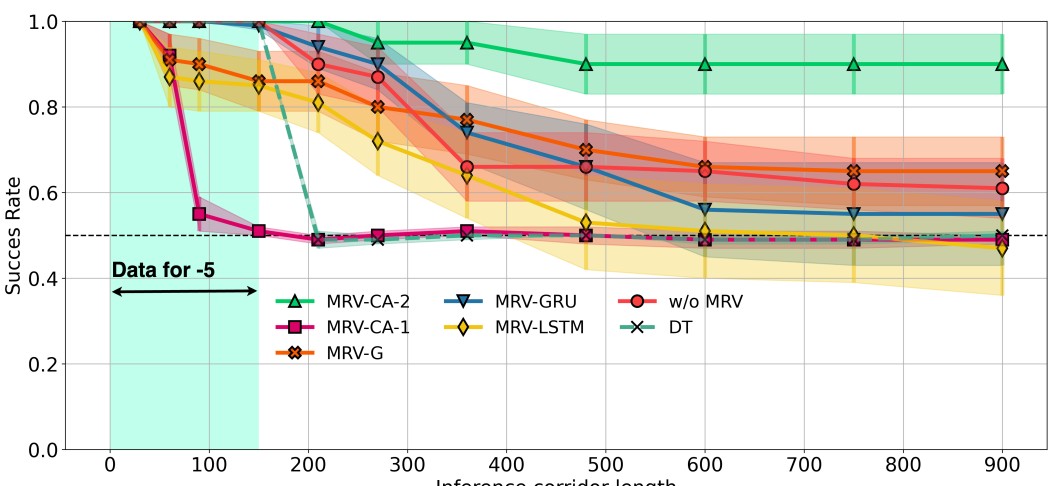

Figure 14: Results of RATE inference with different MRV configurations on the T-Maze environment. Training was performed with the number of segments $N = 5$ and context length $K = 30$, i.e. on trajectories of length $\leq 150$. MRV-CA-2 is the final MRV configuration that is used throughout the work and is designated as MRV.

$$z = \sigma(M_n W_z + M_{n+1} U_z - \texttt{bias}) \tag{3}$$

$$h = \texttt{tanh}(M_n W_g + (M_{n+1} \times r) U_r) \tag{4}$$

$$\tilde{M}_{n+1} = \sigma(M_n(1 - z) + z \times h) \tag{5}$$

The results of the RATE (trained on corridor lengths of $\leq 150$) inference on the T-Maze environment with these MRV configurations are shown in Figure 14 and in Table 6. The results presented in Figure 14 confirm the high stability of RATE when using cross-attention-based MRV (MRV-CA-2), as well as the model's ability to hold important information in memory embeddings when inference on long tasks.

## F.4 ABLATION ON NUMBER OF SEGMENTS AND SEGMENT LENGTH

Partitioning the trajectories into fixed-length segments allows the RATE model to train on long trajectories without increasing the context size, which makes the parameters $N$ (the number of segments into which the training trajectories are divided) and $K$ (the context length, i.e., the size of a single segment) critical because they determine the length of the effective context $K_{eff} = K \times N$. The Figure 15 presents the results of ablation studies for parameters $N$ and $K$ at fixed $K_{eff} = 90$.

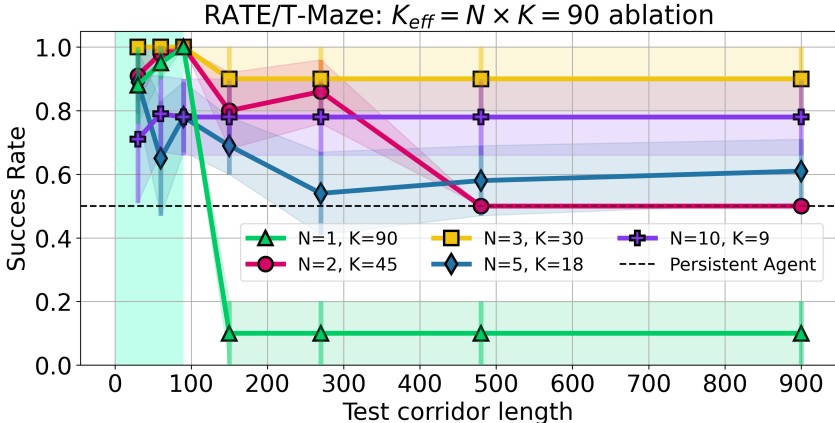

Figure 15: Results of ablation of segments size (context length $K$) and number of segments ($N$) when the effective context length ($K_{eff}$) is fixed: $K_{eff} = K \uparrow \times N \downarrow = 90$.

# G   RECURRENT BASELINES

To prove that all baselines were properly trained and that the results obtained indicate exactly the inability of the considered RNN and SSM baselines to learn on long corridors, we conducted a hyperparameters sweep (see Figure 16).

Results demonstrates that recurrent baselines can solve the T-Maze task when trained on data with moderate corridor lengths (approximately 30 steps, or 90 corresponding tokens) but fail to retain the clue information for longer lengths, unlike a transformer. This is because the transformer's attention mechanism can effectively capture dependencies in highly sparse data, which recurrent models cannot. DT achieves SR$= 0.5$ for $T > K$ for any $K$, while recurrent networks can achieve SR$> 0.5$ in this setting. RATE combines the strengths of transformers (direct access to information in context) and recurrent networks (hidden states for information retrieval).

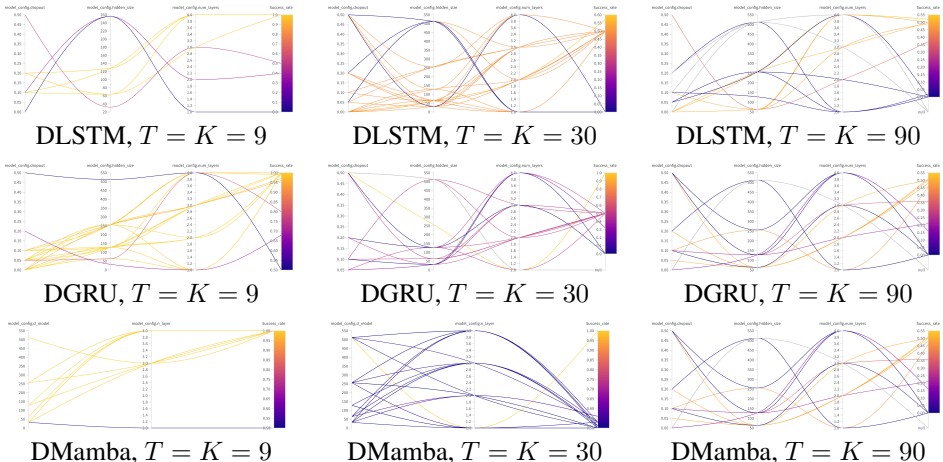

Figure 16: Results of tuning DLSTM, DGRU and DMamba hyperparameters for the T-Maze environment. Validation is performed on corridors of the same length used in training. At each step of the environment, a triplet $(R, o, a)$, i.e., three tokens, is processed.

## H  TRANSFORMER ABLATION STUDIES

**Transformer core hyperparameters.**    This section presents the results of ablation studies on the main hyperparameters of the RATE transformer. The RATE configuration for the T-Maze environment specified in Table 8 was chosen for the ablation studies. The ablation studies focus on understanding the impact of key hyperparameters by systematically varying one parameter while keeping others constant. The results are shown in Figure 17, Figure 18, and Figure 19.

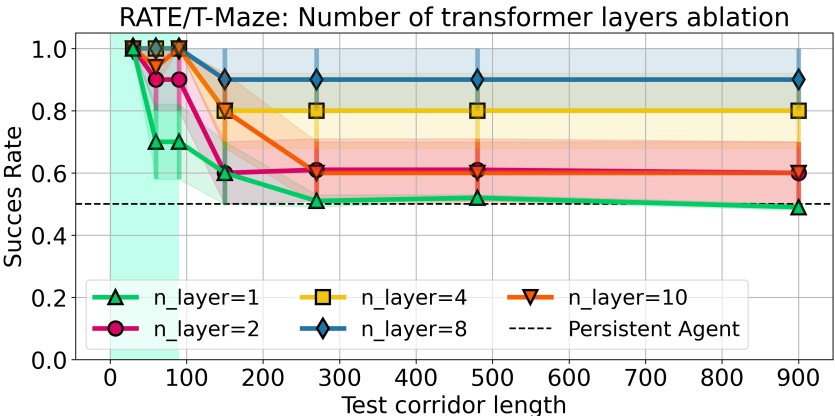

Figure 17: Results of ablation by the number of layers of the RATE model in T-Maze environment.

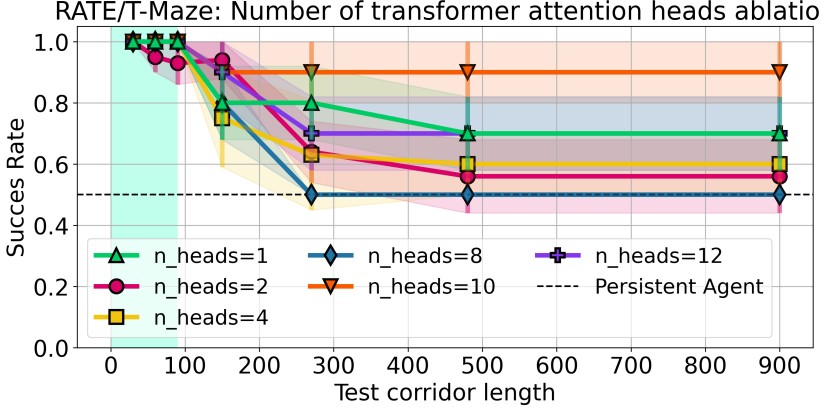

Figure 18: Results of ablation by the number of attention heads of the RATE model in T-Maze environment.

**Feed-Forward Network.**    In our experiments, we found that when the feed-forward network (FFN) is disabled in the transformer decoder, RATE performs slightly better then with FFN enabled. To evaluate the contribution from FFN on the considered baselines, we performed an ablation study on this parameter. The results presented in Figure 20 demonstrate that for RATE alone, disabling FFN gives a performance gain, while the other models' Succes Rate in the T-Maze environment drops.

## I  RECOMMENDATIONS FOR HYPERPARAMETERS SETTINGS

Transformer architectures have many parameters that need to be selected correctly. The use of memory mechanisms in RATE adds a few more hyperparameters. Nevertheless, tuning RATE is

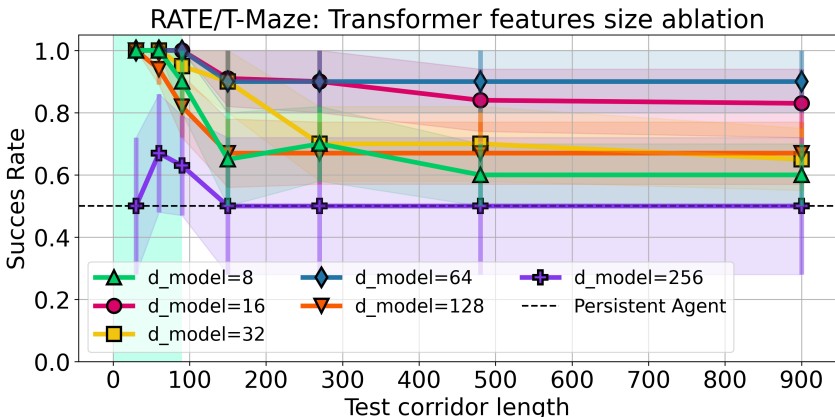

Figure 19: Results of ablation by the features sizes of the RATE model in T-Maze environment.

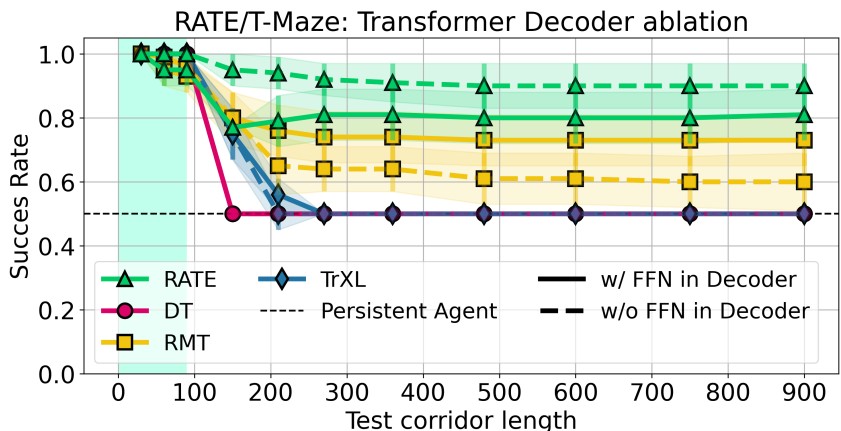

Figure 20: Results of ablation study on use/non-use of feed-forward network in transformer decoder.

practically no different from tuning a regular Transformer. Based on our experience with RATE on different tasks, in this section we provide practical guidelines that aim to simplify the hyperparameters selection process.

To configure the RATE, we recommend performing the setup in the following order:

1. Set the number of segments $N$ into which the trajectory is divided equal to three: $N = 3$. Next, if the length of the trajectories is $T$, set the context length $K = T//3$.

2. Set the default hyperparameters for core transformer and set the following initial parameters related to memory mechanisms:

   (a) Number of memory tokens: `num_mem_tokens` $= 5$

   (b) Number of MRV attention heads: `n_head_ca` $= 2$

   (c) MRV activation function `mrv_act` $= \text{'}relu\text{'}$

   (d) Number of cached hidden states of previous tokens: `mem_len` $= (3 \times K + 2 \times$ `num_mem_tokens` $\times N$ for environments with dense reward function (like ViZDoom-Two-Colors or Minigrid-Memory) or `mem_len` $= 0$ for environments with sparse reward function (like T-Maze).

3. Tune the hyperparameters of the core transformer (number of layers, number of heads, etc.).

4. Adjust the hyperparameters related to the RATE memory mechanisms.

The configuration of the RATE memory mechanisms specified in item (2) of this instruction worked well on all the tasks we considered.

## J    TECHNICAL DETAILS

The Table 11 shows the technical parameters of the training models. Note that the difference between the number of DT and RATE parameters is small and equal to $\delta p = $ d_model $\times$ num_mem_tokens $\sim 10^3$. Training RATE with trajectory splitting into $N$ segments allows $\sim N$ smaller GPU memory size usage than for DT. The training was conducted using a single NVIDIA A100 80 Gb graphics card.

Table 11: Technical configurations of model training. The values in the table are for single run. The training was conducted on a single NVIDIA A100 GPU.

| Env. | Model | GPU mem. | Train time | # params. |
|------|-------|----------|-----------|-----------|
| ViZDoom-Two-Colors | RATE | 24Gb | 4h | 6.0M |
|  | DT | 37Gb |  |  |
| T-Maze | RATE | 5Gb | 2h | 2.4M |
|  | DT | 15Gb |  |  |
| Memory Maze | RATE | 24Gb | 12h | 6.0M |
|  | DT | 37Gb |  |  |
| Minigrid-Memory | RATE | 6Gb | 10h | 6.0M |
|  | DT | 15Gb |  |  |
| Atari | RATE | 21Gb | 9h | 4.7M |
|  | DT | 32Gb |  |  |
| MuJoCo | RATE | 15Gb | 10h | 0.6M |
|  | DT | 45Gb |  |  |

## K    ATTENTION MAPS

In this section, we present attention maps for DT and RATE models in the T-Maze environment in two configurations: $T = K = 15$ (Figure 21, Figure 22) and $T = K = 90$ (Figure 23 and Figure 24). As can be seen from the presented attention maps, DT have attention heads that explicitly define dependencies between the action at the junction and the cue at the beginning of the episode (Head 0, Head 1 in Figure 22). RATE, on the other hand, does not show such dependencies explicitly, but some heads clearly show heavy use of memory tokens (Head 2, Head 4, Head 7 in Figure 21).

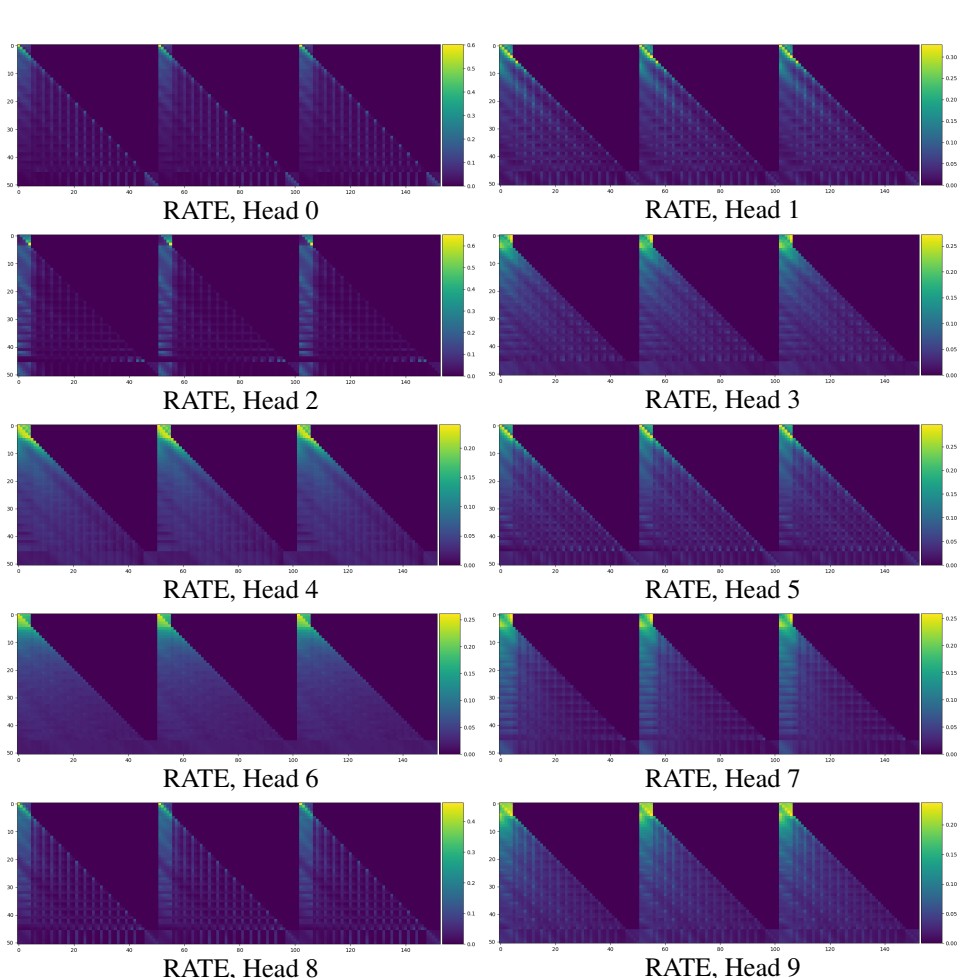

Figure 21: RATE attention maps in the T-Maze environment, $T = K_{eff} = 3 \times 5 = 15$.

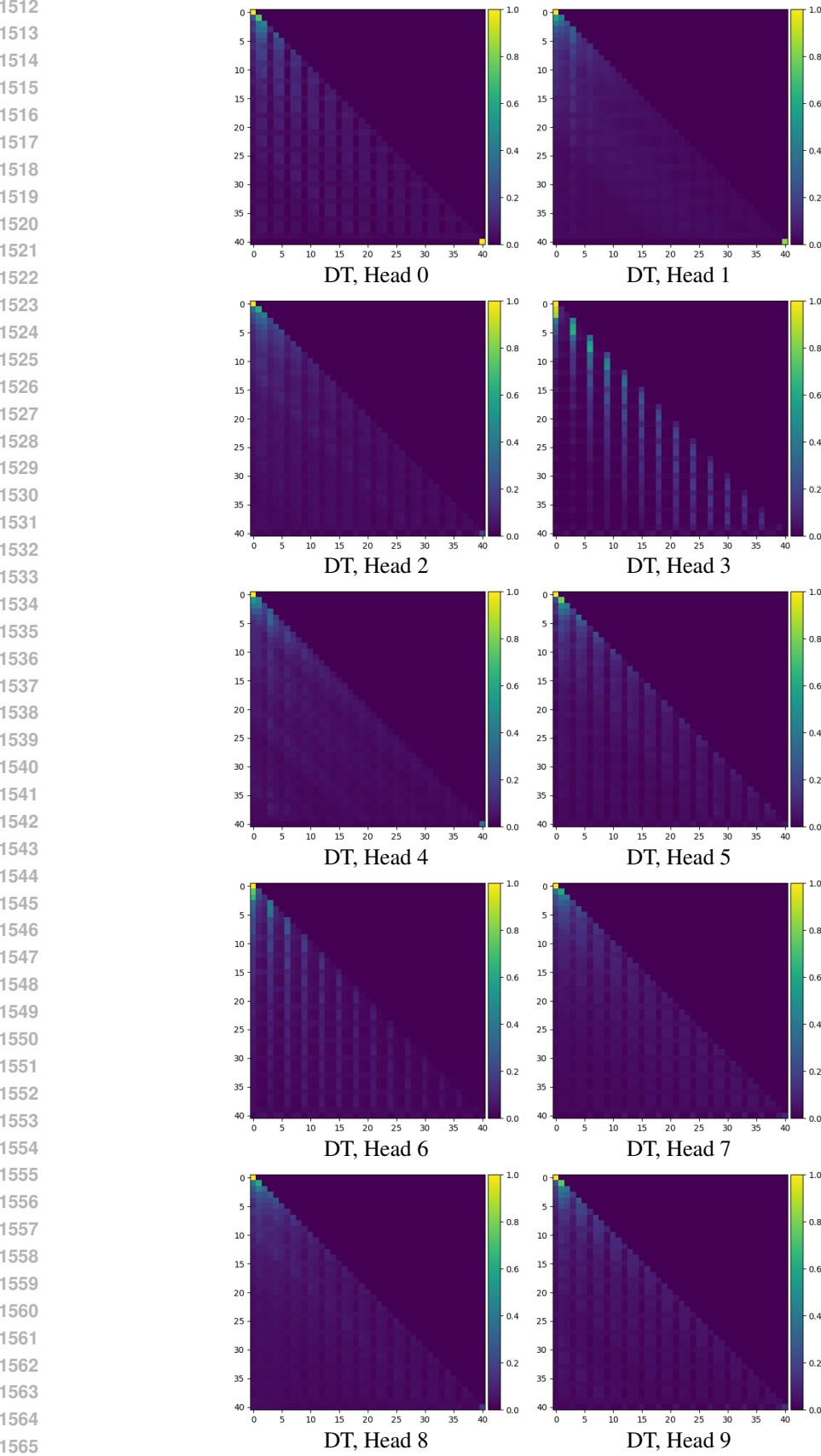

Figure 22: DT attention maps in the T-Maze environment, $T = K = 15$.

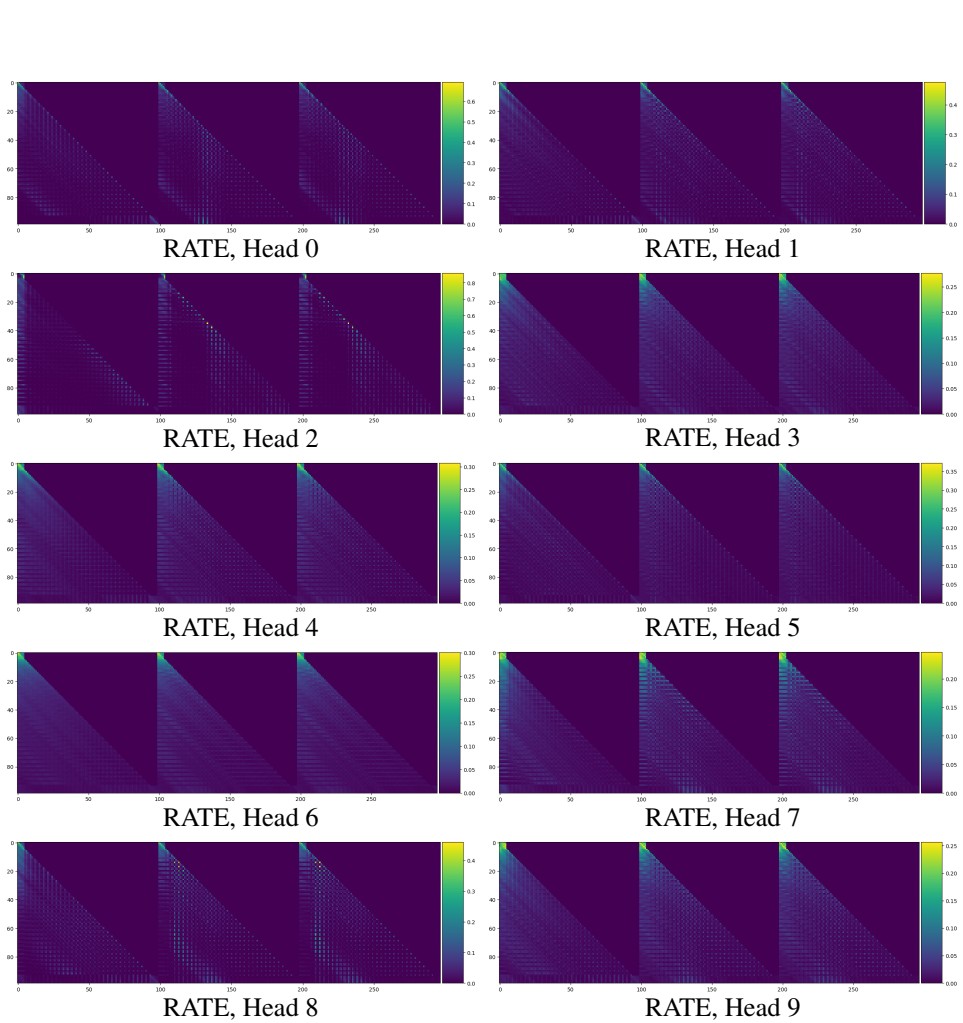

Figure 23: RATE attention maps in the T-Maze environment, $T = K_{eff} = 3 \times 30 = 90$.

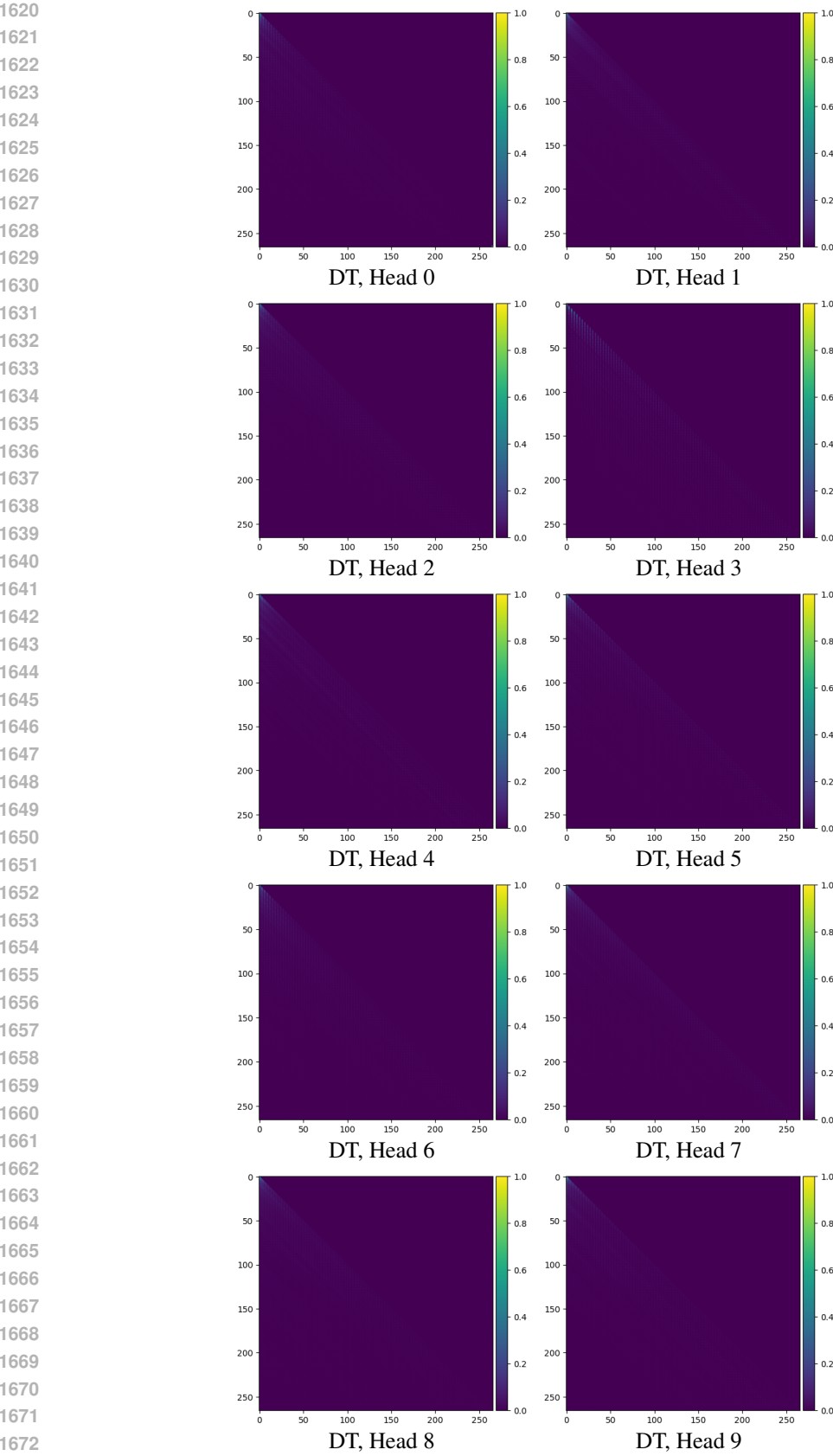

Figure 24: DT attention maps in the T-Maze environment, $T = K = 90$.

