# OpenReview forum: "Recurrent Action Transformer with Memory"
_ICLR.cc/2025/Conference — Submitted to ICLR 2025_

### Official Review · Reviewer_d7jC · 2024-10-21

**Soundness:** 2
**Presentation:** 4
**Contribution:** 2
**Rating:** 5
**Confidence:** 4

**Summary:**

This work investigates the use of memory mechanisms for the Decision Transformer (DT) architecture, a formulation of the offline RL problem as sequence modeling that leverages a transformer network. The motivation is that the vanilla DT does not present an explicit memory mechanism, and the context length is limited due to the quadratic complexity of the attention operation. To address this limitation, the paper proposes RATE, an architecture that conditions the DT architecture with a learned memory embedding. This memory is updated via a Multi-Head Attention module (namely Memory Retention Valve). This module ensures that past sub trajectories can affect future ones, extending the effective context window of the model. The work provides empirical evaluation in several classic and memory-intensive environments, presenting results that are comparable or superior to DT and other memory-augmented architectures.

**Strengths:**

- The work is well-written and easy to follow.

- The methodology looks simple yet effective; while the use of multi-head attention modules for memory is already present in the literature [1], the use for decision transformers looks novel (to the best of my knowledge).

- While there are some concerns in the evaluation setup (see below), the number of experiments is extensive and covers diverse environments. The MRV ablation (RQ3) is particularly interesting.

**Weaknesses:**

- The major concern regarding the work is that the presented empirical evidence does not justify the motivation of straightforwardly assuming the sequence modeling framework for decision-making. It is unclear how this performs in comparison with the standard offline RL or behavior cloning (with simple memory architectures) for memory-intensive environments. For instance, all presented baselines (DT, TrXL, RMT, DLSTM, DGRU, SSM) assume the sequence modeling framework, which is naturally memory demanding. It would be important to contextualize with other simpler architectures, to understand what the gains are of assuming such a modeling framework. It is unclear if sequence modeling is actually needed/helpful for the considered benchmarks. For instance, presenting results of simple LSTM with CQL and (filtered) behavior cloning (as in the DT paper [2], but for memory-intensive environments) would help clarify this.

- The evaluation setup from VizDoom-Two-Colors is questionable. The work leverages a pre-collected dataset that apparently is biased towards one of the colors. The bias does not look slight as it greatly affects the performance of DT, for instance. The choice of considering this dataset ended up considerably affecting the evaluation, as all experiments were required to control for the color. A much simpler solution would be to train a new policy in the environment and ensure this effect does not happen. Why not take this alternative path? The biased data puts in question if the proposed method is indeed performing better than the baselines or if this result is particular to this configuration.

- Appendix D describes that RATE is trained without the feed-forward network because this leads to better results, but it is unclear why that is the case. Also, it is unclear if this same design choice would lead the other baselines to better results. There is no ablation on that, and this choice makes comparisons against the other methods not completely fair.

- The data collection procedure for the Memory Maze (Appendix D) looks arbitrary. Why does the work change the initial sampling timestep following the described rule?
    - Also, the sampling horizons for Atari and MuJoCo are much shorter than the standard for the benchmarks (which is often set to at least 200). This choice of short horizons raises the question if RATE can perform well in the classic setup of these benchmarks, which are commonly handled by non-transformer architectures.


Typo:

L148: embeddingsx -> embeddings

**Questions:**

- From Appendix B.1, it looks like the data collection policies are optimal or near-optimal. Is that the case? It is unclear from the description if that is indeed the case. If that is the case, it would be interesting to know how the proposed method performs with suboptimal data.

- The memory update creates a recursive relationship throughout different trajectory segments. During learning, this should lead to some form of backpropagation through time. How does RATE deal with this? Does it truncate the gradient to flow through past forward passes?

- In Figure 4, why is the performance of TrXL-3 so different from other memory-augmented architectures? Most experiments present similar results among them.


**Summary of the Review**:

The work is clear, easy to follow, and provides a diverse set of environments to show the gains of the proposed method. Nonetheless, there are some design choices in the evaluation setup that are unclear and raise questions about its soundness. Most importantly, the work should bring some simple but crucial baselines to justify and contextualize its motivation of investigating memory in RL as sequence modeling. I believe these are the points that need to be addressed in order to recommend acceptance.

**References**:

[1] Ritter et. al. Rapid Task-Solving in Novel Environments, 2020.

[2] Chen et. al. Decision Transformer: Reinforcement Learning via Sequence Modeling, 2021.

---

> ### Author Response · Authors · 2024-11-23
> **Response to Reviewer d7jC**
>
> Thank you for your feedback. We have responded to your questions and comments and highlighted the changes in the text of the paper in blue.
>
> **W1. Sequence modeling.** Thank you for your thoughtful concern. We would like to clarify several points:
>
> Our work builds upon the sequential modeling framework established in the DT paper, which provides comprehensive experiments demonstrating the benefits of transformer architectures over standard offline RL approaches. The DT paper includes experiments with the Key-to-Door environment, which is memory-intensive as it requires the agent to memorize and utilize specific information after a long sequence of steps.
>
> Our primary contribution focuses on enhancing existing transformer architectures with effective memory mechanisms rather than comparing different learning paradigms (Offline RL vs. BC). The advantage of using sequential modeling with transformers over LSTM architectures is empirically supported by our experiments. Specifically, we demonstrate that LSTMs struggle to propagate gradients beyond approximately 100 steps in a simple T-Maze task, even when trained on optimal trajectories. This limitation is consistent with findings from the NTM paper [1]. Consequently, when action decision at step 100 depend on information from step 0, LSTM-based approaches fail, while our RATE algorithm successfully maintains and utilizes this long-term information and works in both simple and memory-intensive environments.
>
> This fundamental architectural advantage in handling long-term dependencies justifies our focus on improving transformer-based sequential modeling rather than exploring alternative learning frameworks.
>
> **W2. ViZDoom-Two-Colors bias.**
> Memory mechanisms can combat such biases in the dataset (see Fig 3), so we felt it was not necessary to obtain a dataset balanced not only by pillar colors but also by object colors. In addition, when conducting experiments on T-Maze, the dataset for which is perfectly balanced, we obtained that for out-of-context inference DT, unlike baselines with memory, always acts as a persistent agent (the agent turns only up or only down).
>
> **W3. FFN in Decoder.**
> For clarity, we performed additional ablation studies on the use of FFNs in the decoders of the considered transformer baselines (see Fig 20). Disabling FFN allows to obtain better results only for RATE, for other baselines the results deteriorate. Therefore, for fair comparison we disable FFN only for RATE and use FFN for DT, RMT and TrXL.
>
> **W4. Memory Maze data collection.**
> In our dataset, the average total reward per trajectory is <r> ~ 7 with episode length T = 1000. Thus, the agent on average receives a reward +1 for 143 steps. Since we train on three consecutive segments of length 30, we cover a total of 90 environment steps. Thus, to ensure that the reward signal falls within the trajectory segment used for training, we introduce the condition presented in Appendix D. Moreover, we can sample segments from an arbitrary segment for this task.
>
> **W5. Atari and MuJoCo sampling horizons.**
> If by sampling horizons you mean the length of an episode, we have the same length as in DT. In this case, we use only a part of the trajectory during training. Thus, in the original DT paper, a subsequence of length $K=50$ for Pong and $K=30$ for Breakout, Qbert, Seaquest is randomly sampled. In our paper, we sample a subsequence of length $K=30\times 3 = 90$ for RATE. If this does not answer your comment, please clarify what you meant.
>
> **Q1. Data collection policies.**
> We use the optimal policy only for T-Maze. In other memory-intensive tasks, the policies used for trajectories collection are far from optimal.
>
> Since we work in a sequential modeling paradigm, data quality is critical to us, so we use optimal or near optimal policies to collect the data (see Tab 4).
>
> However, as mentioned earlier about ViZDoom, the memory mechanisms allow us to deal with bias in the data, as can be seen in Fig. 3 (c, d), where at t $\in$ [0, 90) (i.e. pillar inside the context) all baselines perform almost the same, but at t $\in$ [90, 189) only the DT result slips, since the pillar is no longer available in the context and the agent starts reproducing bias in the data.
>
> **Q2. BPTT.**
> During training, we use BPTT to propagate gradients through memory tokens across previous segments, which gradients are not stopped between segments. The number of segments for backpropagation is equal to $N$, and ablation results for this hyperparameter are shown in Fig. 15.
>
> **Q3. TrXL performance in T-Maze.**
> On T-Maze, TrXL performs worse than other memory-enhanced baselines because its memory mechanism (caching previous hidden states) mixes representations, preventing the explicit extraction of 1 bit of information. Ablation studies of RATE show that for sparse tasks like T-Maze, this memory mechanism degrades performance.
>
> [1] Graves, A. (2014). Neural Turing Machines. arXiv preprint arXiv:1410.5401.

---

> > ### Comment · Reviewer_d7jC · 2024-11-29
> > **Thank you for your response**
> >
> > Dear authors,
> >
> > Thank you for your rebuttal. After carefully reading it, I understand that my concerns were partially addressed. More specifically, I believe my questions were addressed, and the weaknesses W3/W5 too. Nonetheless, after the clarifications, my concerns related to W1/W2/W4. Still remain.
> >
> > For W1, I still believe it is necessary to contextualize with prior offline RL methods with memory architectures. DT-based methods are very memory demanding, and given the focus on memory tasks, it is unclear if established from RL literature aren't a better alternative. The experiments in the DT paper, while considering Key-to-door, does not consider memory-based architectures for the offline RL algorithms, which is not fully convincing.
> >
> > For W2, I understand that my concern still holds, and the use of the biased data affects the evaluation. Lastly, W4 does look very environment-specific, which raises questions about broader applicability/soundness of the method.
> >
> > For these reasons, I decided to keep my score.

---

> > > ### Author Response · Authors · 2024-12-02
> > >
> > > We thank the reviewer for their insightful comments and have conducted additional experiments to address the concerns raised.
> > >
> > > **W1**
> > > We performed an ablation study comparing the performance of CQL with a LSTM architecture against standard BC in the ViZDoom-Two-Colors environment. Our findings indicate that CQL with LSTM outperforms standard BC with LSTM by 10% (22.18 vs. 20.19) and also surpasses the sequential modeling DLSTM (22.18 vs. 13.1). However, both methods achieve results approximately two times lower than the RATE baseline (45.1).
> > >
> > > |              | CQL+LSTM | BC+LSTM | DT   | RATE | DLSTM |
> > > | ------------ | -------- | ------- | ---- | ---- | ----- |
> > > | Total reward | 22.18    | 20.19   | 24.8 | 45.1 | 13.1  |
> > >
> > > It is important to note that the DLSTM's lower reward is attributed to the extended sequence length, as it requires processing actions and reward-to-go in addition to the state. The LSTM architecture struggles with such long contexts, whereas our proposed RATE effectively leverages the benefits of sequential modeling.
> > >
> > > We hope this additional evidence addresses the reviewer's concerns and demonstrates the **value of sequence modeling frameworks in our benchmarks**. We believe these insights contribute to a more comprehensive understanding of the trade-offs involved in selecting appropriate architectures for decision-making tasks.
> > >
> > > **W2**
> > > The bias toward green objects is due to the environment, not the agent. Thus, **this bias is not present in the first 90 steps** of the trajectories used for training. Specifically, the rewards are R_{reds} = 4.49 and R_{greens} = 4.42, a **difference of slightly over 1%**. However, during inference, due to the way objects are generated in the environment (green objects 'saturate' the environment faster than red ones, hence they are easier for the agent to work with), the trajectories show a bias favoring green objects. Despite this, this paper shows that RATE handles bias better than DT (see Figure 3).
> > >
> > > **W4**
> > > In the sequential modeling paradigm, data quality plays a crucial role in achieving strong performance (see DT paper, Table 3). This is also confirmed by our experiments on MuJoCo on datasets with different data quality (see our paper, Table 4). Due to the poor quality of the dataset used to train the baseline models on the Memory Maze environment, we introduced the filtering procedure described previously. To assess the impact of this procedure, **we conducted additional experiments comparing the performance of DT and RATE trained on the unfiltered dataset.** The results show a significant performance drop, with DT achieving an average reward of 5.5 (-20%) and RATE achieving 5.48 (-28%). Such **degradation is expected** in sequence modeling tasks and is also observed when using BC.
> > >
> > > It is important to note that using filtering for Memory Maze acts as a kind of filtering in %BC. Thus, with the use of filtering, the agent is trained on 6375 trajectories (25.5%), while without filtering it is trained on 25000 trajectories (100%).
> > >
> > > |        | DT (100%) | DT (25.5%) | RATE (100%) | RATE (25.5%) |
> > > | ------ | --------- | ---------- | ----------- | ------------ |
> > > | Reward | 6.83±0.51 | 5.48       | 7.64±0.41   | 5.48         |
> > >
> > > We believe that we have answered all your questions and weaknesses, and the **additional experiments we have conducted** should help increase your score.

---

> > > > ### Author Response · Authors · 2024-12-03
> > > > **Notification**
> > > >
> > > > Please be advised that we have added a new response resolving your concerns.

---

### Official Review · Reviewer_RX6H · 2024-10-28

**Soundness:** 2
**Presentation:** 3
**Contribution:** 2
**Rating:** 5
**Confidence:** 3

**Summary:**

This paper presents the Recurrent Action Transformer with Memory (RATE), which extends a transformer model with a recurrent memory mechanism for offline reinforcement learning. RATE utilizes memory embeddings, caching of previous hidden states, and the Memory Retention Valve to help effective decision making which depends on the history information. Experimental results on memory-intensive environments, Atari games, and MuJoco control environments show that RATE is especially effective when the environment requires memory.

**Strengths:**

1.	This paper is well-organized and a lot of details are included.
2.	RATE is validated on various memory-intensive environments. The results demonstrate RATE achieves good performance on different tasks and environments.
3.	The related works are well-studied and discussed.

**Weaknesses:**

1.	The idea of using the transformer for offline reinforcement learning tasks is not novel. At the same time, the motivation for Memory Retention Valve is not well explained.
2.	It seems that the hyperparameters of RATE are specifically tuned for each environment. At the same time, there are a lot of hyperparameters of RATE, which makes the tuning a really heavy workload.
3.	It is unclear that the hyperparameters of the compared baselines are tuned carefully.

**Questions:**

1.	Why could the Memory Retention Valve significantly improve the performance of RATE in memory-intensive environments? Could the authors give some examples or further explanations?
2.	As there are a lot of hyperparameters, is there some guidance to tune each of them?

---

> ### Author Response · Authors · 2024-11-23
> **Response to Reviewer RX6H**
>
> We are pleased that the reviewer noted the good performance of RATE and the robustness of our experimental results. We have answered the reviewer's questions below. Updates are marked in blue in the text of the article.
>
> **W1.**
> **Transformers in Offline RL.** The works proposing the use of transformers in Offline RLs are mainly focused on MDPs. Our study demonstrates that while transformers like DT perform well on MDPs (e.g., Atari, MuJoCo), their naive application in memory-intensive environments results in severe performance degradation (Figure 5; Tables 2, 5), even failing to address memory tasks entirely (Figures 3, 4; Table 1; Figure 9). Introducing memory mechanisms to transformers resolves this issue, though it introduces challenges, such as excessive memory updating, which we address with MRV.
>
> **Memory Retention Valve.** MRV prevents important information from being “washed out” during long recurrent segment processing. As explained in Section 5 (RQ3), the need for MRV arose from experiments on ViZDoom-Two-Colors and T-Maze. These tasks required critical information to be stored in memory embeddings after the first segment and remain unchanged. However, recurrent updates in subsequent segments risked overwriting this information. To address this, we introduced a gating mechanism to control memory updates, with a cross-attention-based design proving most effective.
>
> **W2. Hyperparameters.**
> Good performance of RATE can be achieved by tuning just two additional parameters, despite the fact that, like any transformer-based model, it has many hyperparameters: the number of memory tokens (num_mem_tokens) and the number of MRV attention heads (n_head_ca). The third key parameter, the number of cached hidden states (mem_len) can be predefined based on the type of task.
>
> In order to make it clearer how changing the main hyperparameters of RATE affects performance, we conducted new ablation studies on the additional RATE parameters: number of segments (N), context length (K), and the use of FFN in the decoder (skip_dec_ffn), and classic transformer parameters: number of attention heads (n_heads), number of transformer layers (n_layer), transformer features size (d_model) (see updates in Appendix F and Appendix H).
>
> Ablation studies on num_mem_tokens, mem_len, n_head_ca, N, and K are presented in Appendix F. For environments with continuous rewards (e.g., ViZDoom-Two-Colors), optimal results are achieved when mem_len = (3xK + 2x num_mem_tokens) x N, where  K  is context length and  N  is the number of segments ( K_{eff} = N \times K ). For sparse environments (e.g., T-Maze), mem_len = 0 is most effective.
>
> Other transformer parameters, such as the n_heads, n_layers, d_model, skip_dec_ffn, can be tuned independently of memory hyperparameters. A minimal starting configuration is num_mem_tokens = 5, n_head_ca = 2, and mem_len = 0 or (3xK + 2x num_mem_tokens) x N, depending on trajectory structure (see Table 8, Figures 10, 11). Additional results are provided in Appendix H.
> To clarify hyperparameter tuning, we’ve added guidelines in the Appendix I.
>
> **W3. Baselines tuning.** To ensure a fair comparison, we used the same base transformer hyperparameters for DT, RMT, TrXL, and RATE. For RMT, TrXL, and RATE, we tuned parameters related to memory mechanisms. In Atari and MuJoCo experiments, we used the same transformer parameters as in the original DT paper.
>
> In turn, for DLSTM, DGRU and DMamba tuning we also tuned by basic parameters and reported the results corresponding to the best configuration (see Appendix G). At the request of Reviewer hxTt, we have posted the code and results of tuning these baselines in the `recurrent_baselines/` directory of our code repository, an updated link to which is available in the abstract of updated paper.
>
>
> **Q1. Why could the MRV improve the performance of RATE?**
> Recurrent segment processing in RATE aims to update memory tokens to carry important information between segments. However, when memory tokens are updated for a long time, important information can be lost from them (e.g., on T-Maze we get information about a clue on the first segment, but it can be used for decision making only on segment 30, see Fig 4, Tab 6). To control the updating of information, we need a kind of valve that will keep important information inside the memory tokens, but will not interfere with the writing of new information. We have tried different ways to implement such a valve (Tab 6, Fig 14), and came to the conclusion that the best results can be obtained if we let the model itself determine how to implement the gating of memory tokens. The cross-attention architecture is the best for this purpose, as evidenced by our experiments.
>
> **Q2. Guide how to tune RATE.** We are grateful to you for a really good idea. For ease of use of our model, we have added a section to the appendix (see Appendix I) with our recommendations on how best to tune RATE for different tasks.

---

> > ### Comment · Reviewer_RX6H · 2024-11-26
> >
> > The reviewer appreciates the authors' feedback. I would like to maintain my score.

---

> > > ### Author Response · Authors · 2024-11-26
> > >
> > > Thank you for your reply! **We have answered all your previous questions and comments** and would like to understand **what doubts you still have about our answers** that prevent you from raising your grade to a positive one?
> > >
> > > Moreover, **we have conducted thorough ablation studies** on RATE parameters: on transformer (see Appendix H) and on memory mechanisms (Appendix F). By the way, ablation on memory, which is the most important in our work, **was also present before rebuttal**.
> > >
> > > Furthermore, we **added a detailed guide on how to tune RATE** to get the first positive results in memory-intensive environments (Appendix I). A small part of it **was present in Appendix F before rebuttal**.
> > >
> > > We have even **marked the changes in the text in blue** for your convenience.
> > >
> > > We look forward to hearing your responses and continuing the discussion.

---

> > > > ### Comment · Reviewer_RX6H · 2024-11-26
> > > >
> > > > I keep my score as there are two remaining concerns.
> > > >
> > > > First, there seems to be a lot of hyperparameters in the proposed method and they may differ in different tasks.
> > > >
> > > > Second, the paper needs to make clear denifition about memory-intensive and shows clear evidence that the proposed method can solve it.

---

> ### Author Response · Authors · 2024-11-26
>
> Thank you so much for your prompt response! We really appreciate it.
>
> Regarding your first concern about hyperparameters - **any transformer** architecture, including DT, **contains many hyperparameters**. Our RATE model contains **only 5 more hyperparameters** than DT: `num_mem_tokens`, `mem_len`, `n_head_ca`, `mrv_act`, `sections`. In our initial ablation studies, supplemented by your recommendation, we show that of these 5 additional hyperparameters **only `sections`, i.e. number of segments, should be configured separately**, while the others **already in the initial configuration allow to obtain significantly better results** than just DT (see Appendix F, Appendix H).
>
> We **emphasized the simplicity of RATE tuning** in an additional section in Appendix I. Thus, **RATE can be tuned almost in the same way as DT**, and even with coarse tuning of memory-intensive parameters, it can solve memory-intensive problems.
>
> In your review **in Strengths 2, you noted that “RATE is validated on various memory-intensive environments. The results demonstrate RATE achieves good performance on different tasks and environments.”**
>
> In section 4.1. “memory-intensive environments” **we define this term:** “memory-intensive environments - environments where the agent requires memory to operate successfully”. In other words, these are POMDPs where the agent's decision making in the present depends on information from the past that can be retrieved using memory.
>
> **All our work aims to show that the proposed architecture successfully solves memory-intensive environments** and common tasks. Thus, Fig. 3, Fig. 4, Fig. 5, Tab. 1, Tab. 2, show the benefits of using RATE to solve memory-intensive tasks, while Tab. 3 and Tab. 4 show that RATE is not inferior to SOTA solutions in the Offline RL domain when solving classical problems.
>
> We really **appreciate your quick comments and engagement in the discussion**, and we really hope we can get our RATE proposal to you.

---

> > ### Author Response · Authors · 2024-12-02
> >
> > Dear Reviewer RX6H,
> >
> > We appreciate your time and effort in reviewing our application.
> >
> > We would like to clarify our response regarding the tuning of hyperparameters for RATE. As stated in our paper, RATE was trained using the same hyperparameters as DT, except for those specifically related to memory. The recommended configuration for memory-related hyperparameters (`num_mem_tokens=5`, `n_head_ca=2`, `mrv_act=‘relu’`, `mem_len=(3 x K + 2 x num_mem_tokens) x N` with `sections (N) = 3`) is already **sufficient to outperform DT on memory-intensive tasks** (see Appendix I). Therefore, in the initial iteration, selecting hyperparameters for RATE does not differ significantly from the process for DT.
> >
> > Furthermore, using **environment-specific hyperparameter configurations is entirely natural**, as each environment has unique complexity and characteristics. For example, in the original DT paper, distinct sets of hyperparameters were used for solving Atari and MuJoCo tasks.
> >
> > We also assert that **we have provided clear evidence that our model successfully addresses memory-intensive tasks**. This is **demonstrated in the experimental results across T-Maze, ViZDoom-Two-Colors, Minigrid-Memory, and Memory Maze environments**, where agent performance heavily depends on memory.
> >
> > We hope this reassures you that there is **no cause for concern regarding the number of hyperparameters or the evidence supporting RATE’s effectiveness** in memory-intensive environments.
> >
> > **We have addressed all your questions and comments and conducted extensive additional experiments.** We hope this will positively influence your final evaluation.

---

### Official Review · Reviewer_cJAo · 2024-10-29

**Soundness:** 3
**Presentation:** 3
**Contribution:** 3
**Rating:** 6
**Confidence:** 4

**Summary:**

This paper studies how to design a new transformer architecture for RL tasks where history memory is needed. Due to the quadratic memory complexity of the vanilla transformer, naively expanding the context length would not result in an efficient solution to memory-intensive RL tasks. Therefore, authors propose to store memory information into a chunk that is updated after sequential processing of each segment. Through extensive experiments and detailed ablation study, the chosen memory chunk updating mechanism ---MRV--- is proven effective and superior towards other design choices.

**Strengths:**

## originality:
The usage of memory chunk bears some similarity to existing works but the proposed MRV is  original and interesting, to my knowledge.

## quality:
The method is well supported by its experiments and ablation studies.

## clarity:
The paper is well written. I find most of it easy-to-read.

## significance:
Efficient Transformer for memory-intensive (RL) tasks is of great need and research interest. And this paper successfully designs a novel MRV method for it. Through the experiments conducted, the efficacy and advantage is demonstrated against some baseline methods.

**Weaknesses:**

I have several  concerns regarding this work:
1. lack of theoretical guidance. Since there is no theoretical guidance towards the design of RATE/MRV, I  think they are  likely empirically determined. This somehow shadows the contribution of this work. But I understand demanding this is beyond the scope.
2. ablation study. As Transformer itself is prone to many hyperparameters, I hope to see more ablation studies regarding various components of RATE (details in questions)

**Questions:**

1. what is "cached hidden states"? It is mentioned multiple times throughout the paper but lacks a clear definition.
2. which Positional Encoding (PE) is used? Do you use any PE in MRV module?
3. could you provide some experiments comparing RATE to ERNIE-Docs? The code change should be just minimal from TrXL
4. in your Table 1, the performance of DMamba in T-Maze is quite lower to DT under the same context window (K = 90). Why is this?  What is the corridor length?
5. I hope to see more of ablation studies. Especially about the capacity of memory chunk, the length of segment $S_n$ and the number of layers of transformer.  In appendix F, some of them seem to have been performed but  are correlated  with the "cached hidden states" and I don't quite understand what you're ablating on
6. could you show the  attention map  of causal and cross attention  during evaluation of the method ?

---

> ### Author Response · Authors · 2024-11-23
> **Response to Reviewer cJAo**
>
> Thank you for recognizing our algorithm’s effectiveness, especially for memory-intensive tasks requiring efficient transformers. We value your feedback and have addressed your comments. In the updated version of the text, the changes are highlighted in blue.
>
> **Q1. Cached hidden states:** Cached hidden states refer to saved hidden states from previous tokens (as in TrXL [1]). This mechanism combines current hidden states with cached ones, enabling access to past information. The mem_len parameter sets how many past token states are available for processing each segment. We’ve clarified this in the paper.
>
> **Q2. Positional Encoding:** In our architecture, we employ relative sinusoidal positional embeddings, as in TrXL [1]. This design enables the model to incorporate previously computed hidden states when attending to memory. The approach is invariant to sequence shifts and supports generalization to sequences of varying lengths, which is important in RL.
>
> However, we do not use positional encoding within the MRV block for the following reasons:
> 1. Memory tokens serve as an abstract repository of information, where their content, rather than their position, is of primary importance.
>
> 2. The number of memory tokens is typically small and fixed, making positional encoding less relevant.
>
> 3. The primary function of the MRV mechanism is to update the memory state through interactions among memory tokens, rather than rely on positional relationships.
>
> **Q3. ERNIE-Docs baseline:** Thanks for the suggestion! We're eager to see how ERNIE-Docs' memory mechanisms perform in memory-intensive settings. Since there's no existing implementation for Offline RL, we'll work on it ourselves and aim to share results before the rebuttal period ends.
>
> **Q4. T-Maze and recurrent baselines:** For the T-Maze environment, we everywhere specify the episode duration T through a tight bound on the corridor length L:T=L+2. Because of this constraint, the agent is guaranteed to receive a reward of 0 if it performs a single action outside the optimal policy. We added the performance of a random agent to Table 1 in the attached paper text.
>
> In the T-Maze experiment shown in Table 1, we test the ability of the models to train on trajectories of a given length. This table for T-Maze shows results for models trained on data with trajectories of length T=3-90 (i.e., with corridor lengths L=1-88) and failed only on trajectories of length T=90 (L=88). It is important to note that all considered baselines process a triplet (R,o,a), i.e., 3 tokens, at each step of the environment, and thus trajectories of length T=90 have a length of 3x90=270 tokens.
>
> The results show recurrent baselines fail to retain key information over 270-token episodes, while transformers excel at linking sparse information via attention.
>
> To make the results of this experiment clearer, we performed additional experiments with baseline learning on trajectories of length T=9 and T=30 (with context K=9 and K=30, respectively):
>
> | |DLSTM|DGRU|DMamba|DT|RATE|
> |-|-|-|-|-|-|
> |T=K=9|1.0|1.0|1.0|1.0|1.0|
> |T=K=30|0.6|1.0|1.0|1.0|1.0|
> |T=K=90|0.5|0.5|0.5|1.0|1.0|
>
> The new results confirm the ability of recurrent baselines to capture sparse dependencies on small time horizons (T=30 corresponds to 90 tokens) and their inability to learn on longer sequences.
>
> In the case where only observations instead of triplets are fed at each step, the situation changes slightly:
>
> | |LSTM|GRU|Mamba|
> |-|-|-|-|
> |T=K=9|1.0|1.0|1.0|
> |T=K=30|0.72|1.0|1.0|
> |T=K=90|0.5|1.0|1.0|
> |T=K=270|0.0|0.32|0.55|
>
> Here, unlike in the previous table, T=90 corresponds to 90 tokens rather than 270 tokens. In turn, the recurrent baselines also show an inability to solve the task at T=270 (as in the previous table, when T=90 corresponded to 270 tokens), indicating directly the limited time window available for recurrent baselines to memorize during training.
>
> **Q5. Hyperparameters:** We fully agree with you, and therefore we performed additional ablation studies on the number of transformer layers, the number of attention heads, the model dimensionality, the use of FFN in the decoder, and the segment lengths under a fixed effective context. We have added the results to the updated text of our paper in Section F. Additional Ablation Studies and Section H. Transformer Ablation Studies.
>
>
> The initial results of the ablation studies in Section F referred only to the new RATE parameters, such as number of memory tokens (num_mem_tokens), number of cached hidden states of previous tokens (mem_len), number of MRV's attention heads (n_head_ca), and MRV activation function (mrv_act). We answered about cached hidden states in Q1.
>
> **Q6. Attention maps:** Thanks for the tip! We have added attention maps for RATE and DT to the Appendix, Section K.
>
> [1] Dai, Z. (2019). Transformer-xl: Attentive language models beyond a fixed-length context. arXiv preprint arXiv:1901.02860.

---

### Official Review · Reviewer_hxTt · 2024-10-30

**Soundness:** 1
**Presentation:** 2
**Contribution:** 1
**Rating:** 3
**Confidence:** 3

**Summary:**

The authors propose a memory-augmented transformer to use for long episodes in reinforcement learning. At a high level, their approach augments a decision transformer with a recurrent update to alleviate the memory pressure caused by $O(n^2)$ space complexity. The authors propose different gating mechanisms applied to the recurrent state that they call "valves". They compare their model to other types of decision transformers, and perform an ablation of valves.

**Strengths:**

- The authors introduce a modification to decision transformers that improves performance
- The paper is fairly easy to follow

**Weaknesses:**

I find some of the experimental results dubious, specifically the T-Maze experiments. In particular, the fact that only the author's proposed method and the decision transformer can solve a simple T-maze task. To succeed at the T-maze task, all a sequence model must do is memorize one bit of information (left or right) for 90 timesteps. Given that Mamba and its predecessors S4/S5 have been shown to memorize much more information over 16,000 timesteps gives me pause. 90 timesteps should be well-within the memory capabilities of even the LSTM or GRU. Furthermore, a score of 0.5 is what I would expect a memory-free agent to achieve, and this is precisely the score that the Mamba/LSTM/GRU models achieve.

I suspected the authors may have made a mistake during implementation, so I examined the "T-Maze_new" directory provided in the anonymous code link. However, there is no code or references to Mamba, GRU, or LSTM, and so I am not sure how the authors arrived at their conclusion. I do not want to sound overly harsh, but reproducibility is the biggest issue facing RL today.

Furthermore, most of the experiment tasks are what I would consider poor tests of memory. As far as I understand, both T-Maze and VizDoom require storing 1 bit of information presented at the initial timestep, for a duration of less than 100 timesteps. So I would not consider this a sufficient comparison between RATE and other related work.

Finally, I do not think this paper is very novel. Using a decision transformer to solve a reinforcement learning task is certainly not novel, and there have been numerous papers that extend the context length of transformers through the use of a recurrent state. I would consider this an incremental work, which would be fine if the experiments were convincing. But they are not.

**Questions:**

- Why are LSTM, GRU, Mamba unable to make *any* progress beyond on a random agent on such a simple task?
    - Where is the code for these baselines?
- Why did you create your own valve/gating mechanism instead of using one of the existing methods proposed by prior recurrent transformers?

---

> ### Author Response · Authors · 2024-11-22
> **Response to Reviewer hxTt**
>
> Thank you for your valuable feedback. We have addressed your comments in the revised text, highlighting the changes in blue, and look forward to further discussion.
>
> ### **Reproducibility (W1-2, Q2)**
> We agree that reproducibility is crucial in RL, and you are right to emphasize its importance. While we cited the open-source implementations of DLSTM [1] and DMamba [2], we’ve **updated our repository** with a recurrent_baselines/ section, **including code for all recurrent baselines** and instructions for running them in T-Maze and ViZDoom-Two-Colors. The anonymized repository link can be found in the abstract of the updated version of the paper.
>
> ### **Environments (W3)**
> In our work, we consider not only T-Maze (which is a **common memory test in RL** and has been used in numerous papers, e.g. [3-6], to name a few) and ViZDoom-Two-Colors, but also other challenging environments such as Minigrid-Memory and Memory Maze, on which we perform extensive experiments (Fig 5, Tab 2).
>
> Even the T-Maze environment is more complex than it seems, as the agent receives unique information only at the first and the last steps of an episode. The **random agent gets SR= 0** starting from the corridor length L = 5. Since the episode length is defined as T=L+2 and any deviation from the optimal policy leads to failure (we have added the random agent results to Tab 1 and Fig 4). **SR=0.5 corresponds to a persistent agent** that successfully reaches the junction, but then makes a turn up or down randomly.
>
> Speaking of episode length, we represent each step of the environment as a triplet of tokens (R, o, a), so that when **processing a context of length K, we actually process 3K tokens**:
> In T-Maze, we trained agents on trajectories of length L = 3 - 90 and validated on trajectories of length L = 3 - 900. Thus, at L = 900, the agent demonstrates information memorization at 900 steps and successfully **processes 2700 tokens**, well in excess of 100 steps.
> In ViZDoom-Two-Colors, the RATE agent trained on the first 90 steps of the environment then operates for ~1700 steps (episode timeout = 2100), **processing ~1700 × 3 = 5100 tokens** — again, far exceeding 100 steps.
>
> ### **LSTM, GRU, Mamba, and Transformer (W1, Q1-2)**
> In Tab 1, we show results in the T-Maze environment for baselines trained on trajectories of 90 steps and validated on corridors of length 90. As noted earlier, 90 steps correspond to 270 tokens.
>
> To prove that **all baselines were properly trained**, we included the code for LSTM, GRU, and Mamba and conducted a hyperparameter sweep (see App G - Recurrent Baselines), as per your request. To confirm that these models can solve the T-Maze task in principle but fail on long sequences, we added results for T=K={9, 30, 90}. Results for the best configurations are shown below:
> | |DLSTM|DGRU|DMamba|DT|RATE|
> |------|-----|----|------|---|----|
> |T=K=9|1.0|1.0|1.0|1.0|1.0|
> |T=K=30|0.6|1.0|1.0|1.0|1.0|
> |T=K=90|0.5|0.5|0.5|1.0|1.0|
>
> This demonstrates that recurrent baselines can solve the T-Maze task when trained on data with moderate corridor lengths but fail to retain the clue information for longer lengths, unlike a Transformer. This is because the transformer's attention mechanism can effectively capture dependencies in highly sparse data, which recurrent models cannot. DT achieves SR = 0.5 for T>K for any K, while recurrent networks can achieve SR > 0.5 in this setting. RATE combines the strengths of transformers (direct access to information in context) and recurrent networks (hidden states for information retrieval).
>
> ### **Novelty (W4, Q3)**
> Sequential updates in our model’s memory tokens can cause information loss, as shown in the T-Maze task (Tab 6). To address this, we introduce a memory valve (MRV) to regulate memory token updates.
> New valve mechanism (MRV) is exactly **one of the contributions of our work**. If you are referring to GTrXL [7] as "recurrent transformers with gating mechanisms", we tested their approach (MRV-G in our paper), which underperformed compared to our method (Tab 6). If not, we would appreciate **references to other works on recurrent transformers with gating mechanisms for Offline RL tasks** that you mentioned to support our future research.
>
> [1] Siebenborn, M. et al. How crucial is transformer in decision transformer?NeurIPS 2022
>
> [2] Ota, T. Decision mamba: Reinforcement learning via sequence modeling with selective state spaces.arXiv:2403.19925.
>
> [3] Ni, T. et al. When do transformers shine in rl? decoupling memory from credit assignment.ICLR  2024
>
> [4] Grigsby, J. et al. Amago: Scalable in-context reinforcement learning for adaptive agents.ICLR 2024
>
> [5] Esslinger, K. et al.. Deep transformer q-networks for partially observable reinforcement learning. arXiv:2206.01078.
>
> [6] Pramanik, S. et al. AGaLiTe: Approximate Gated Linear Transformers for Online Reinforcement Learning.TMLR 2024
>
> [7] Parisotto, E. et al. Stabilizing transformers for reinforcement learning.ICML 2020

---

> > ### Author Response · Authors · 2024-12-02
> >
> > Dear Reviewer hxTt!
> >
> > We would like to thank you once again for your time and effort spent on our submission, as well as for the feedback provided.
> >
> > We have answered all your questions and comments and performed additional experiments as requested, and now we would like to know if you have any additional questions or concerns?

---

> > > ### Author Response · Authors · 2024-12-04
> > > **Final comment**
> > >
> > > We would like to point out that we have carefully answered all questions and concerns of the Reviewer hxTt, performed extensive additional experiments to substantiate our claims, and provided code for recurrent baselines. Despite these efforts, the Reviewer hxTt did not continue the discussion or provide any follow-up responses. Since we have responded to all questions and comments, we believe there is no reason not to raise our score.
> > >
> > > Highlights of our response to Reviewer hxTt:
> > > 1. Proved that ViZDoom-Two-Colors and T-Maze are good tasks for memory testing
> > > 2. Showed why exactly transformers can solve T-Maze (on large corridor lengths), while recurrent transformers cannot
> > > 3. Conducted additional experiments on recurrent baselines, which show that they can solve T-Maze, but on significantly smaller corridors than transformers
> > > 4. Posted the code for recurrent baselines and performed a sweep to prove that everything works correctly
> > > 5. Argued the motivation and novelty of our work

---

### Author Response · Authors · 2024-12-04
**General Response**

We thank the reviewers for their valuable feedback, highlighting the strengths of RATE, including its efficiency (Reviewer hxTt, Rx6H, d7jC), simplicity (Reviewer d7jC), originality in memory processing (Reviewer cJAo), and extensive experiments (Reviewer cJAo, RX6H) .

The motivation for RATE stems from the limitations of classical Offline RL methods (e.g., CQL, BC) and recurrent baselines (LSTM, GRU, Mamba) in solving memory-intensive tasks over long horizons. Transformer-based models (DT) are also restricted to tasks fitting entirely within their context. RATE overcomes these challenges, as shown in environments like ViZDoom-Two-Colors and T-Maze, as well as outperforms other transformers with memory.

Our novelty lies in leveraging transformer with memory for Offline RL and introducing the Memory Retention Valve, which prevents forgetting during long inferences, outperforming existing alternatives.

Since we have answered all the questions and concerns of the hxTt reviewer, as well as performed additional experiments and posted the code for additional recurrent baselines, the hxTt reviewer has ignored further discussion, we believe this should contribute to our score.
We would like to address the following points of our response to reviewer hxTt separately.
1. Proved that ViZDoom-Two-Colors and T-Maze are good tasks for memory testing
2. Showed why exactly transformers can solve T-Maze (on large corridor lengths), while recurrent transformers cannot
3. Conducted additional experiments on recurrent baselines, which show that they can solve T-Maze, but on significantly smaller corridors than transformers
4. Posted the code for recurrent baselines and performed a sweep to prove that everything works correctly
5. Argued the motivation and novelty of our work

Overall, to eliminate reviewers' concerns, in addition to answering all questions in detail, we conducted additional experiments:
1. We added DLSTM, DGRU, DMamba, DT, RATE training results on T-Maze environment with T=K=9 and T=K=30 configurations using triplets (R, o, a) at each step of the environment
2. Added LSTM, GRU and Mamba training results on T-Maze environment with configurations T=K=9, T=K=30, T=K=90 and T=K=270 using only observations at each step of the environment
3. Conducted additional ablation studies on the main parameters of the transformer: `n_layers`, `n_heads`, `d_model`.
4. Conducted additional ablation studies on the parameters related to memory mechanisms: `skip_ffn_dec`, `sections (N)`, `context length (K)`.
5. Conducted experiments with CQL+LSTM and BC+LSTM on ViZDoom-Two-Colors environment, demonstrating the benefits of using sequence modeling framework for decision-making in memory-intensive environemnts
6. Conducted experiments with %BC on Memory Maze, in conjunction with results for MuJoCo, confirming the importance of data quality when working in a sequence modeling framework
7. Performed sweeps of DLSTM, DGRU and DMamba hyperparameters to prove that these models are only able to solve T-Maze when trained on episodes of length less than T=90, unlike RATE.
8. (Optional) Added code for DLSTM, DGRU and DMamba experiments

We have performed a large number of additional experiments and added their results to the Appendix of our paper. We hope that the work we have done will contribute to the acceptance of our work.

Regards,
RATE authors

---

### Meta-Review · Area_Chair_uPdy · 2024-12-19

**Metareview:**

This paper introduces RATE, a recurrent transformer architecture with memory mechanisms designed for offline RL tasks. While the reviewers acknowledged the paper's thorough empirical evaluation and potential benefits in memory-intensive environments, several critical concerns remain unresolved. First, although the authors provided extensive ablation studies and implementation details in their response, the work lacks sufficient theoretical justification for the proposed memory retention mechanism. Second, the evaluation on non-standard variants of existing memory tasks (e.g., biased VizDoom dataset, arbitrary filtering in Memory Maze) raises questions about the broader applicability and robustness of the approach. Most importantly, the paper fails to convincingly demonstrate advantages over simpler memory architectures (e.g., LSTM with standard offline RL algorithms) in memory-intensive settings - the additional experiments comparing against CQL+LSTM show limited improvements that may not justify the increased architectural complexity.

Given these fundamental limitations in theoretical foundation and experimental validation, I recommend rejection.

**Additional Comments On Reviewer Discussion:**

While the reviewers acknowledged the paper's thorough empirical evaluation and potential benefits in memory-intensive environments, several critical concerns remain unresolved. First, although the authors provided extensive ablation studies and implementation details in their response, the work lacks sufficient theoretical justification for the proposed memory retention mechanism. Second, the evaluation on non-standard variants of existing memory tasks (e.g., biased VizDoom dataset, arbitrary filtering in Memory Maze) raises questions about the broader applicability and robustness of the approach. Most importantly, the paper fails to convincingly demonstrate advantages over simpler memory architectures (e.g., LSTM with standard offline RL algorithms) in memory-intensive settings - the additional experiments comparing against CQL+LSTM show limited improvements that may not justify the increased architectural complexity.

---

### Decision · Program_Chairs · 2025-01-22

Reject